# A Practical Descent Method for Singular Value Decomposition

## Abstract

Singular Value Decomposition (SVD) is a long-established technique, with most existing methods relying on matrix-based formulations. However, matrix operations are relatively less friendly to parallelization and distributed computation compared to descent-based methods, motivating the need for alternative approaches. Descent-based methods offer a promising direction, yet existing ones such as Riemannian gradient descent suffer from inefficiency due to the need for repeated projections onto nonlinear manifolds. In this work, we introduce a novel descent method for SVD grounded in a primal–dual reformulation. Specifically, we construct a least-squares primal problem whose dual corresponds to the SVD. We show that (i) the non-zero KKT solutions of the primal problem yield the singular vectors of the matrix, and (ii) inexact singular value estimation still ensures bounded reconstruction error. Building on these results, we propose an iterative descent-based algorithm, Des-SVD, along with scalable variants leveraging random sampling and parallelization. Extensive experiments demonstrate that Des-SVD achieves significantly higher computational efficiency compared to prior descent methods, while remaining competitive with matrix-based algorithms. Our implementation is publicly available at: `https://anonymous.4open.science/r/Descent-SVD-method`.

## 1 Introduction

Singular Value Decomposition (SVD) is a fundamental and important technique in linear algebra, extensively applied to diverse fields. Along with the explosive application of computer vision (Rajwade et al., 2013; Guo et al., 2016; Kumar & Vaish, 2017; Yang & Lu, 1995) and natural language processing (Meng et al., 2024), the size of the matrices involved in SVD problems is steadily increasing, which emphasizes the urgent need for more efficient SVD methods.

However, these matrix-based methods face challenges in parallelization and still require centralized computation on the server (Chai et al., 2024). Descent methods offer an alternative (Qian, 1999; Jain et al., 2018; Chen et al., 2020), being well suited for parallel computing (Richtárik & Takáč, 2016; Liu et al., 2022; Bai et al., 2024) and stochastic sampling (Martino et al., 2018; Luengo et al., 2020; Akyildiz & Míguez, 2021). Yet a practical descent method for SVD is still lacking. The existing Riemannian gradient method (Sato & Iwai, 2013), for example, is hampered by costly manifold projections.

A pioneering work by Suykens (2016) introduces a least squares problem and demonstrates that SVD satisfies its Karush-Kuhn-Tucker (KKT) conditions, thereby opening the door for the development of descent methods for SVD. However, the primal-dual relationship faces a key obstacle since the least squares problem is non-convex. Thus, while singular values and vectors can form a local optimum, a local optimum does not directly yield the exact SVD.

Building on (Suykens, 2016), we establish a practical path from a local optimum of the least squares problem to the SVD in this paper. Our method (Des-SVD) is available for parallelization and random sampling, and

we believe that additional speed-up methods could be developed in the future. Experimental results show that our method is far more efficient than the Riemannian gradient method (Sato & Iwai, 2013; Sato, 2021) and achieves comparable performance to the matrix-based methods (Menon & Elkan, 2011; Feng et al., 2018; Gao et al., 2025) in experiments involving images and large matrices.

## 2 SVD AND ITS LEAST SQUARES FORMULATION

Let us first review Singular Value Decomposition (SVD). For a matrix $\boldsymbol{A} \in \mathbb{R}^{n \times m}$, the SVD factorizes $\boldsymbol{A}$ into the product of three matrices:

$$\boldsymbol{A} = \boldsymbol{U}\boldsymbol{\Sigma}\boldsymbol{V}^{\top}, \tag{1}$$

where $\boldsymbol{U}$ is an $n \times n$ orthogonal matrix, $\boldsymbol{V}$ is an $m \times m$ orthogonal matrix, and $\boldsymbol{\Sigma}$ is an $n \times m$ diagonal matrix containing the non-negative singular values of $\boldsymbol{A}$ on its diagonal.

Because both $\boldsymbol{U}$ and $\boldsymbol{V}$ are orthogonal matrices, i.e., $\boldsymbol{U}^{\top}\boldsymbol{U} = \boldsymbol{I}_n$ and $\boldsymbol{V}^{\top}\boldsymbol{V} = \boldsymbol{I}_m$, we can rewrite the SVD equation as follows:

$$\begin{aligned} \boldsymbol{A}\boldsymbol{V} &= \boldsymbol{U}\boldsymbol{\Sigma}, \\ \boldsymbol{A}^{\top}\boldsymbol{U} &= \boldsymbol{V}\boldsymbol{\Sigma}. \end{aligned} \tag{2}$$

This Lanczos Decomposition Theorem forms the basis of the Lanczos algorithm (Lanczos, 1958).

A topic closely related to SVD is eigen-decomposition, for which there is also a desire to develop descent methods to speed up the process. Here, we list some interesting papers for reference (Tisseur, 2001; Knyazev, 2001; Marek et al., 2014; Ogita & Aishima, 2018). However, these methods all rely on symmetry or even positive semi-definiteness, which are not applicable to SVD, as it is a decomposition for non-square matrices.

The foundation of our work is given by Suykens (2016), which treats SVD as a dual of a least squares problem, specifically, a variant LS-SVM (Suykens & Vandewalle, 1999). The central idea of the work lies in defining two feature mappings of the matrix $\boldsymbol{A}$ as follows:

$$\begin{cases} \varphi(\boldsymbol{x}_i) = \boldsymbol{D}^{\top}\boldsymbol{x}_i, \\ \psi(\boldsymbol{y}_j) = \boldsymbol{y}_j, \end{cases} \tag{3}$$

where $\boldsymbol{x}_i$ and $\boldsymbol{y}_j$ are the $i^{\text{th}}$ row vector and the $j^{\text{th}}$ column vector of $\boldsymbol{A}$ respectively, and $\boldsymbol{D}$ is a compatible matrix satisfying $\boldsymbol{A}\boldsymbol{D}\boldsymbol{A} = \boldsymbol{A}$.

This group of feature mappings establishes a primal-dual relationship between a least squares problem (primal) and the SVD (dual). By setting $\gamma = 1/s$, where $s$ is a singular value of $\boldsymbol{A}$, we obtain the following primal formulation for the corresponding pair of singular vectors:

$$\min_{\boldsymbol{w},\boldsymbol{v},\boldsymbol{e},\boldsymbol{r}} J(\boldsymbol{w},\boldsymbol{v},\boldsymbol{e},\boldsymbol{r}) = -\boldsymbol{w}^{\top}\boldsymbol{v} + \frac{1}{2}\gamma\sum_{i=1}^{N}e_i^2 + \frac{1}{2}\gamma\sum_{j=1}^{M}r_j^2$$

$$\text{s.t.} \quad e_i = \boldsymbol{w}^{\top}\varphi(\boldsymbol{x}_i),\ i = 1,\dots,n,$$
$$r_j = \boldsymbol{v}^{\top}\psi(\boldsymbol{y}_j),\ j = 1,\dots,m, \tag{4}$$

where $\boldsymbol{w}, \boldsymbol{v} \in \mathbb{R}^n$ and $e_i, r_j \in \mathbb{R}$. Let $[\boldsymbol{\alpha}]$ and $[\boldsymbol{\beta}]$ represent the complete dual solutions corresponding to a series of primal problems, where the singular values of $\boldsymbol{A}$ are considered respectively. The key idea is that if $[\boldsymbol{\alpha}]$ and $[\boldsymbol{\beta}]$ are the SVD solutions of $\boldsymbol{A}$, they must satisfy the KKT conditions of (4), which are shown below:

$$\begin{aligned} \boldsymbol{A}[\boldsymbol{\beta}] &= [\boldsymbol{\alpha}]\boldsymbol{\Sigma}, \\ \boldsymbol{A}^{\top}[\boldsymbol{\alpha}] &= [\boldsymbol{\beta}]\boldsymbol{\Sigma}. \end{aligned} \tag{5}$$

Another key property is that the target value in the primal problem converges to zero when the dual variables align with the singular vectors, providing a clear convergence criterion for gradient descent. The detailed proof is shown in Appendix A.

## 3 SVD FROM PRIMAL SPACE

The above pioneering work demonstrates a new avenue for developing descent methods for SVD. However, eq. (4) is a non-convex problem, meaning that different local optima can lead to different dual solutions, with SVD being just one of them. In other words, the existing discussion indicates that SVD can satisfy the KKT condition for eq. (4), but it cannot guarantee that solving eq. (4) will necessarily yield an SVD. This section will address this fundamental obstacle step by step: (1) we will prove that when a singular value is given and the regularization coefficient is set accordingly, the descent method can lead to the singular vector by normalizing the non-zero solutions of the KKT condition; (2) we will explain the reason why the descent method will lead to zero solutions for the KKT condition when the regularization coefficient is incorrect; (3) we will prove that when a small error is tolerated, an inexact estimation of the singular value is sufficient to obtain the singular vectors, which yields lower cost than the exact computation.

### 3.1 FROM KKT TO SINGULAR VECTOR

As mentioned earlier, while an SVD solution satisfies the KKT condition eq. (5), the reverse is not necessarily true. In this section, we will demonstrate that, when $\gamma$ is chosen as the reciprocal of the singular value $s$, any **non-zero** point that satisfies the KKT condition can be normalized to yield the corresponding singular vector.

Let us start from the Lagrangian of eq. (4):

$$\mathcal{L}(\boldsymbol{w}, \boldsymbol{v}, \boldsymbol{e}, \boldsymbol{r}; \boldsymbol{\alpha}, \boldsymbol{\beta}) = J(\boldsymbol{w}, \boldsymbol{v}, \boldsymbol{e}, \boldsymbol{r}) - \sum_i \alpha_i \left( e_i - \boldsymbol{w}^\top \varphi(\boldsymbol{x}_i) \right) - \sum_j \beta_j \left( r_j - \boldsymbol{v}^\top \psi(\boldsymbol{y}_j) \right). \tag{6}$$

The Karush–Kuhn–Tucker conditions imply that

$$\begin{cases} \frac{\partial \mathcal{L}}{\partial \boldsymbol{w}} = 0 \implies \boldsymbol{v} = \sum_i \alpha_i \varphi(\boldsymbol{x}_i), \\ \frac{\partial \mathcal{L}}{\partial \boldsymbol{v}} = 0 \implies \boldsymbol{w} = \sum_j \beta_j \psi(\boldsymbol{y}_j), \\ \frac{\partial \mathcal{L}}{\partial e_i} = 0 \implies \gamma e_i = \alpha_i, \ \forall i, \\ \frac{\partial \mathcal{L}}{\partial r_j} = 0 \implies \gamma r_j = \beta_j, \ \forall j, \\ \frac{\partial \mathcal{L}}{\partial \alpha_i} = 0 \implies e_i = \boldsymbol{w}^\top \varphi(\boldsymbol{x}_i), \ \forall i, \\ \frac{\partial \mathcal{L}}{\partial \beta_j} = 0 \implies r_j = \boldsymbol{v}^\top \psi(\boldsymbol{y}_j), \ \forall j. \end{cases} \tag{7}$$

As noted in Section 2, we define the dual variables as $[\boldsymbol{\alpha}] = [\boldsymbol{\alpha}_1, \ldots, \boldsymbol{\alpha}_n]^\top$ and $[\boldsymbol{\beta}] = [\boldsymbol{\beta}_1, \ldots, \boldsymbol{\beta}_m]^\top$, where each dual pair $(\boldsymbol{\alpha}_k, \boldsymbol{\beta}_k)$ is associated with a singular value $s_k$ and its corresponding target problem eq. (4). Therefore, the stacked vectors must satisfy the orthogonality conditions $[\boldsymbol{\alpha}]^\top [\boldsymbol{\alpha}] = \boldsymbol{I}_n$ and $[\boldsymbol{\beta}]^\top [\boldsymbol{\beta}] = \boldsymbol{I}_m$, which are not explicitly enforced by the KKT condition eq. (7). Building on this trivial observation, we will demonstrate that any non-zero solution to the KKT condition eq. (7) can be transformed into the corresponding singular vector through data normalization.

We first prove the natural orthogonality of $[\boldsymbol{\alpha}]$ and $[\boldsymbol{\beta}]$. If we only consider the column vector of the dual variables $\boldsymbol{\alpha}_k$ and $\boldsymbol{\beta}_k$, we can rewrite eq. (2) as

$$\boldsymbol{A}\boldsymbol{\alpha}_k = \lambda_k \boldsymbol{\beta}_k, \tag{8}$$

$$\boldsymbol{A}^\top \boldsymbol{\beta}_k = \lambda_k \boldsymbol{\alpha}_k. \tag{9}$$

Left-multiplying both sides of the equation in eq. (9) by matrix $\boldsymbol{A}$, we obtain:

$$\boldsymbol{A}\boldsymbol{A}^\top \boldsymbol{\beta}_k = \lambda_k \boldsymbol{A}\boldsymbol{\alpha}_k. \tag{10}$$

Substituting the expression from eq. (8) into the above equation, we get:

$$\boldsymbol{A}\boldsymbol{A}^\top \boldsymbol{\beta}_k = \lambda_k^2 \boldsymbol{\beta}_k. \tag{11}$$

We can conclude that $\boldsymbol{\beta}_k$ is one of the singular vectors of the normal matrix $\boldsymbol{A}\boldsymbol{A}^\top$. According to the property of normal matrices, the singular vectors corresponding to different singular values of the normal matrix are orthogonal (Golub & Van Loan, 2013). Therefore, we can easily prove that the columns of $[\boldsymbol{\beta}]$ satisfy orthogonality, and the proof for $[\boldsymbol{\alpha}]$ can be done in the same way.

In addition to orthogonality, the normalization property is also satisfied. For each vector $\boldsymbol{\alpha}_i$ in the matrix $[\boldsymbol{\alpha}] = [\boldsymbol{\alpha}_1, \ldots, \boldsymbol{\alpha}_n]^\top$, the constraints of the KKT equation eq. (7) hold, revealing a constant-ratio relationship between $\boldsymbol{\alpha}_k$ and the corresponding element $\boldsymbol{e}_k$. Similarly, an analogous relationship exists between each vector $\boldsymbol{\beta}_r$ in the matrix $[\boldsymbol{\beta}] = [\boldsymbol{\beta}_1, \ldots, \boldsymbol{\beta}_m]^\top$ and the corresponding element $\boldsymbol{v}_r$.

As a result, the normalization property of the matrices $[\boldsymbol{\alpha}]$ and $[\boldsymbol{\beta}]$ can be obtained by normalizing the columns of the matrices $\boldsymbol{E} = [\boldsymbol{e}_1, \ldots, \boldsymbol{e}_n]^\top$ and $\boldsymbol{V} = [\boldsymbol{v}_1, \ldots, \boldsymbol{v}_m]^\top$ respectively. Since normalization ensures orthogonality without affecting the KKT conditions eq. (7), we implement normalization only after all iterations are completed.

**Remark 1.** *For the non-zero KKT solutions of eq. (4), orthogonality is naturally satisfied due to the implicit constraints in the KKT conditions, which stem from the properties of normal matrices.*

### 3.2 FEASIBLE DESCENT DIRECTION TO NON-ZERO SOLUTION

The key to finding the singular vectors by solving the target problem eq. (4) is identifying a vector that satisfies the KKT condition in eq. (5). It is crucial to set $\gamma$ as the reciprocal of a singular value $s$ for this condition to hold. If this requirement is not met, equation eq. (5) cannot be satisfied by any non-zero vectors. To illustrate this, we will examine the practical algorithm and demonstrate that a feasible descent direction leads to a zero solution for eq. (5).

We stack the primal variables as $\boldsymbol{x} := [\boldsymbol{w}, \boldsymbol{v}, \boldsymbol{e}, \boldsymbol{r}]$ for notational convenience. We first show that $\Delta\boldsymbol{x} = -\boldsymbol{x}$ is a feasible direction at the initial step. Next, we prove that when $\gamma \neq 1/s$, the KKT matrix is full-rank. Together, these two points imply that $\Delta\boldsymbol{x} = -\boldsymbol{x}$ is the only feasible descent direction for any $\boldsymbol{x}$. Therefore, the update $\boldsymbol{x}^1 = \boldsymbol{x}^0 - t\Delta\boldsymbol{x}$ converges to zero.

To start with, let us consider the constraint matrix $\boldsymbol{C}$ and the Hessian matrix $\boldsymbol{H}$ of eq. (4):

$$\boldsymbol{C} = \begin{bmatrix} \boldsymbol{\Phi} & \boldsymbol{0} & \\ \boldsymbol{0} & \boldsymbol{\Psi} & -\boldsymbol{I}_{e,r} \end{bmatrix} \in \mathbb{R}^{(m+n)\times(3n+m)}, \tag{12}$$

$$\boldsymbol{H} = \begin{bmatrix} \boldsymbol{0} & -\boldsymbol{I}_w & \boldsymbol{0} \\ -\boldsymbol{I}_v & \boldsymbol{0} & \\ \boldsymbol{0} & & \gamma\boldsymbol{I}_{e,r} \end{bmatrix} \in \mathbb{R}^{(3n+m)\times(3n+m)}, \tag{13}$$

where $\boldsymbol{\Phi} = [\varphi(\boldsymbol{x}_1); \varphi(\boldsymbol{x}_2); \cdots ; \varphi(\boldsymbol{x}_n)] \in \mathbb{R}^{n\times n}$ and $\boldsymbol{\Psi} = [\psi(\boldsymbol{y}_1); \psi(\boldsymbol{y}_2); \cdots ; \psi(\boldsymbol{y}_m)] \in \mathbb{R}^{m\times n}$.

We know that the constraint matrix $\boldsymbol{C}$ satisfies the following equations:

$$\boldsymbol{C}\boldsymbol{x} = \boldsymbol{0} \Leftrightarrow \begin{bmatrix} \boldsymbol{\Phi} & \boldsymbol{0} & \\ \boldsymbol{0} & \boldsymbol{\Psi} & -\boldsymbol{I}_{e,r} \end{bmatrix} \begin{bmatrix} \boldsymbol{w} \\ \boldsymbol{v} \\ \boldsymbol{e} \\ \boldsymbol{r} \end{bmatrix} = \boldsymbol{0} \Leftrightarrow \begin{cases} e_i = \boldsymbol{w}^\top \varphi(\boldsymbol{x}_i), \ \forall i, \\ r_j = \boldsymbol{v}^\top \psi(\boldsymbol{y}_j), \ \forall j. \end{cases} \tag{14}$$

Now we consider the KKT matrix:

$$\boldsymbol{K} = \begin{bmatrix} \boldsymbol{H} & \boldsymbol{C}^\top \\ \boldsymbol{C} & \boldsymbol{0} \end{bmatrix} \in \mathbb{R}^{(4n+2m)\times(4n+2m)}, \tag{15}$$

and the KKT function:

$$\begin{bmatrix} H & C^\top \\ C & 0 \end{bmatrix} \begin{bmatrix} \Delta x \\ \Delta v \end{bmatrix} = - \begin{bmatrix} g + C^\top v \\ Cx \end{bmatrix}, \tag{16}$$

where $v$ is the Lagrange operator initialized as $0$, and $g$ is the gradient of the target function. If we focus solely on the descent direction $\Delta x$ and set $\Delta v = 0$ at the **first** step, we can convert the original KKT matrix into two functions:

$$\begin{cases} H\Delta x = -g, \\ C\Delta x = -Cx. \end{cases} \tag{17}$$

On the other hand, the value of $Hx$ can be calculated:

$$Hx = [-v, -w, \gamma e, \gamma r], \tag{18}$$

which is simply the gradient of the target problem, so we have $Hx = g$, i.e., $H(-x) = -g$. Consequently, it follows that $\Delta x = -x$ is a feasible solution to eq. (17). Furthermore, we will demonstrate the properties of the KKT matrix $K$ in Theorem 3.1, and its proof is shown in Appendix B.1.

**Theorem 3.1.** $\forall s \in \mathbb{R}$, *if $s$ is not a correct singular value of the matrix $A$, the KKT matrix $K$ remains full rank, which implies that $\Delta x = -x$ is the unique solution to the KKT function eq.* (16). *Conversely, if $s$ is a correct singular value, there exists at least one non-zero solution to eq.* (16).

From Theorem 3.1, we know that if $\gamma$ is chosen incorrectly, $K$ will become full rank, causing $x^1$ to become zero after the first update step. This proves the necessity of setting the correct singular value $s$ theoretically.

### 3.3 FAST ESTIMATION OF INEXACT SINGULAR VALUES

The above fact seemingly suggests that only when an accurate singular value is provided can a descent method be used to solve SVD accurately. However, in practice, exact singular values cannot be obtained due to numerical errors. For the same reason, one cannot expect to exact SVD; equivalently, it is not necessary to precisely fit the KKT matrix. Suppose we tolerate errors within $\varepsilon_1$ for row reconstruction and $\varepsilon_2$ for column reconstruction, respectively. The following theorem discusses the corresponding requirements on the accuracy of singular value estimation. Its proof is given in Appendix B.2.

**Theorem 3.2.** *Let $s$ be the true singular value and $\gamma \in \left[ \frac{1}{s} - \Delta\gamma, \frac{1}{s} + \Delta\gamma \right]$. Suppose that $\Delta\gamma$ satisfies the following condition:*

$$\|\Delta\gamma\| < T_{\mathrm{err}} \triangleq \min \left\{ \frac{\varepsilon_1}{\left( \sum_i \|D^\top x_i\|^2 \right)^{\frac{1}{2}}}, \frac{\varepsilon_2}{\left( \sum_j \|y_j\|^2 \right)^{\frac{1}{2}}} \right\}. \tag{19}$$

*Then, there exists a feasible descent direction to non-zero solutions for the KKT conditions, within the approximation tolerances $\varepsilon_1$ and $\varepsilon_2$. Specifically, the following conditions hold:*

$$\begin{cases} \|v - \sum_i \gamma e_i \varphi(x_i)\| & < \varepsilon_1, \\ \|w - \sum_j \gamma r_j \psi(y_j)\| & < \varepsilon_2. \end{cases} \tag{20}$$

As a result, fast algorithms for singular value estimation with an error less than $T_{\mathrm{err}}$ become applicable. We apply the Rayleigh quotient iteration (Rajendran, 2002; Simoncini & Eldén, 2002) because of its accuracy and efficiency by finishing the estimation without calculating the full SVD. The detailed method is provided in Appendix D for reference.

# 4 DESCENT ALGORITHM FOR SVD

## 4.1 DESCENT METHOD FOR A GIVEN SINGULAR VALUE

The theoretical framework presented enables us to propose a descent method for SVD in the primal space. The proposed algorithm is divided into two primary steps: estimating the singular values and applying the descent method to compute the singular vectors.

As an inexact singular value $s$ could be efficiently estimated, we suppose it has been obtained and focus on the descent method for solving equation eq. (4) to compute the corresponding singular vector. The connection between the primal least squares problem and the SVD is established through the KKT condition, which imposes strict feasibility requirements on equation eq. (4). Meanwhile, by applying random sampling, the matrix size is not large, which allows us to choose Newton's method (Chen et al., 2020). A failure-detection and auto-restart mechanism is also implemented. It reports failure and initiates a restart once the variable nears zero concurrently or the objective value turning negative. The algorithm details are provided in Algorithm 1.

---

**Algorithm 1** Descent method for calculating the singular vectors from a given singular value.

---

**Input:** $\boldsymbol{A} \in \mathbb{R}^{n \times m}$ : the target matrix ; $\boldsymbol{C} \in \mathbb{R}^{(m+n) \times (3n+m)}$ : the coefficient matrix of equality constraints s.t. $\boldsymbol{C}\boldsymbol{x} = \boldsymbol{0}$ ; $\gamma$: the reciprocal of the given singular value $s$; $n_{\max}$: the max iterations for the newton method; $\varepsilon$: the threshold for the convergence;
**Output:** $\boldsymbol{\alpha}, \boldsymbol{\beta}$: the corresponding singular vectors of $s$.
1: Initialize the primal variable $\boldsymbol{x} \in \mathbb{R}^{3n+m}$ .
2: Do the variable mapping $\boldsymbol{w} = \boldsymbol{x}[: n], \boldsymbol{v} = \boldsymbol{x}[n : 2n], \boldsymbol{e} = \boldsymbol{x}[2n : 3n], \boldsymbol{r} = \boldsymbol{x}[3n :]$.
3: Construct the loss function in eq. (4).
4: **for** $i = 1$ to $n_{\max}$ and $J > \varepsilon$ **do**
5:      **if** $\|\boldsymbol{x}\| < 1 \times 10^{-10}$ or $J < -1$. **then**
6:          Report failure and start the auto-restart mechanism
7:      **end if**
8:      Calculate $\mathcal{L}$'s Hessian matrix $\boldsymbol{H}$ in eq. (13).
9:      Get $\Delta\boldsymbol{x}$ by solving $\begin{bmatrix} \boldsymbol{H} & \boldsymbol{C}^\top \\ \boldsymbol{C} & \boldsymbol{0} \end{bmatrix} \begin{bmatrix} \Delta\boldsymbol{x} \\ \Delta\boldsymbol{v} \end{bmatrix} = - \begin{bmatrix} \boldsymbol{g} + \boldsymbol{C}^\top \boldsymbol{v} \\ \boldsymbol{C}\boldsymbol{x} \end{bmatrix}$ .
10:      Use line search to update $\boldsymbol{x}$.
11: **end for**
12: Get the normalized dual variables $\boldsymbol{\alpha} = \frac{\boldsymbol{e}}{\|\boldsymbol{e}\|} \in \mathbb{R}^n$ and $\boldsymbol{\beta} = \frac{\boldsymbol{v}}{\|\boldsymbol{v}\|} \in \mathbb{R}^m$.
13: **return** $\boldsymbol{\alpha}, \boldsymbol{\beta}$

---

## 4.2 THE REFINED DESCENT SVD ALGORITHM

Since each singular value is computed independently, Des-SVD naturally supports parallelization, with minimal communication required as only the singular vectors are gathered in the final stage. It can also be accelerated through random sampling; in particular, randomized SVD (Halko et al., 2011) constructs a matrix $\boldsymbol{Q}$ with $k = k(\varepsilon)$ orthonormal columns approximating the subspace of $\boldsymbol{A}$, satisfying $\|\boldsymbol{A} - \boldsymbol{Q}\boldsymbol{Q}^*\boldsymbol{A}\| \le \varepsilon_c$, where $\varepsilon_c$ denotes the computational tolerance.

After parallelization and randomized sampling, the KKT system for an $m \times n$ matrix with $k$ singular values reduces directly from $4n + 2m$ to $6k$. Therefore, the overall complexity of Des-SVD is $O((6k)^3)$, which is of the same magnitude as the classical Lanczos method with complexity $O(k^3)$.

The overall Des-SVD algorithm is summarized in Appendix C, supporting both parallelization and random sampling. Experiments are conducted with `parallel = True` and `randomized = True`.

## 5 EXPERIMENTS

To evaluate the accuracy and efficiency of our Des-SVD, we conduct experiments on images and random matrices. The baseline methods include the Riemannian gradient method (Sato & Iwai, 2013) (referred to as Rie-SVD), the standard randomized SVD algorithm (Halko et al., 2011), which applies Jacobi SVD after dimensionality reduction (referred to as Jac-SVD), and the Lanczos method with dimensionality reduction (referred to as Lan-SVD). The comparison between Des-SVD and Rie-SVD will show significant improvement in computational efficiency over other descent-based methods. The comparison to Jac-SVD and Lan-SVD will verify that Des-SVD is comparable to Lan-SVD and faster than Jac-SVD.

To ensure fairness, all methods are manually implemented without relying on pre-existing library functions and tested on the same CPU resources. Each experiment is repeated 10 times for statistical validity. The hyperparameter settings of all four methods and the ablations of Des-SVD are shown in Appendix E. We evaluate both time and accuracy across varying singular values. The SVD accuracy is defined as $R_{\mathrm{acc}} = 1 - \frac{||USV^{\top} - A||_{\mathrm{F}}}{||A||_{\mathrm{F}}}$, where $A$ is the target matrix, and $U, S, V^{\top}$ are the SVD components of $A$.

### 5.1 SVD ON LOW-RANK MATRICES

We compare the four methods on low-rank matrices and report their performance in Table 1, showing mean values as variances are negligible. Rie-SVD is the slowest due to repeated projections and convergence difficulty on larger sizes. In contrast, Des-SVD formulates SVD as a parallelizable least squares problem via the primal dual relationship, achieving computational efficiency comparable to classical matrix-based methods for the first time.

**Table 1:** Performance Comparison of Jac-SVD, Lan-SVD, Rie-SVD, and Des-SVD on low-rank matrices.

| m, n, k | Jac-SVD | | Lan-SVD | | Rie-SVD | | Des-SVD (Ours) | |
|---|---|---|---|---|---|---|---|---|
| | $R_{\mathrm{acc}} \uparrow$ | Time $\downarrow$ | $R_{\mathrm{acc}} \uparrow$ | Time $\downarrow$ | $R_{\mathrm{acc}} \uparrow$ | Time $\downarrow$ | $R_{\mathrm{acc}} \uparrow$ | Time $\downarrow$ |
| 30, 10, 2 | 20.01% | 0.01s | 20.38% | 0.04s | 20.37% | 1.05s (Iter 145) | **20.38%** | **0.01s** |
| 30, 10, 5 | 54.03% | 0.02s | 54.04% | 0.05s | 54.04% | 42.20s (Iter 7370) | **54.04%** | **0.02s** |
| 300, 10, 5 | 64.65% | 0.06s | 64.66% | 0.05s | 64.65% | 32.88s (Iter 5295) | **64.66%** | **0.02s** |
| 300, 20, 10 | 71.05% | 0.05s | 71.05% | 0.05s | 64.76% | 56.49s (Iter 7630) | **71.05%** | **0.04s** |

### 5.2 SVD ON GRAYSCALE IMAGES

Next, we evaluate Des-SVD on image data, a key application area of SVD. As noted above, Rie-SVD is less efficient; hence, the following experiments focus on Des-SVD and two representative matrix-based methods. We randomly sample 25 grayscale images of size $1024 \times 1024$ from the FFHQ dataset[1]. The reconstruction performance of the selected images (*PeppersRGB* [2], *Cat* and *Church*[3]) using Des-SVD and Jac-SVD is shown in Figures 1 - 3. In general, these methods yield similar accuracy but differ in computational time. Therefore, we omit the accuracy and report only the computational time in Table 2.

### 5.3 SVD ON RANDOM MATRICES

Beyond image processing, SVD is also relevant in many more general applications. To test this, we generate synthetic matrices of larger size $10000 \times 10000$. Since all methods achieve similar accuracy, we focus on time consumption, reported in Table 3.

---

[1] https://github.com/synctrust/ffhq-dataset.git
[2] https://www.eecs.qmul.ac.uk/~phao/IP/Images/
[3] https://www.pexels.com

**Table 2:** Performance Comparison of Different SVD Methods on FFHQ Dataset.

| k | Lan-SVD | Jac-SVD | Des-SVD(Ours) |
|---|---------|---------|---------------|
| 10 | $0.076 \pm 0.001$s | $\mathbf{0.021 \pm 0.003}$s | $0.059 \pm 0.000$s |
| 20 | $0.082 \pm 0.004$s | $0.073 \pm 0.014$s | $\mathbf{0.065 \pm 0.000}$s |
| 50 | $0.224 \pm 0.003$s | $0.625 \pm 0.093$s | $\mathbf{0.144 \pm 0.000}$s |
| 100 | $\mathbf{0.244 \pm 0.033}$s | $2.999 \pm 0.199$s | $0.322 \pm 0.000$s |

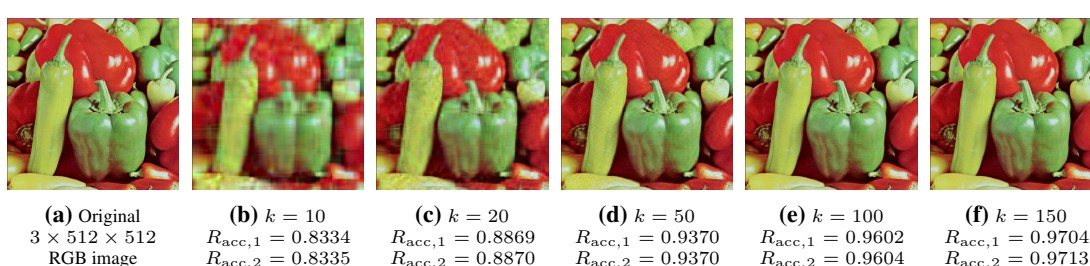

**(a)** Original $3 \times 512 \times 512$ RGB image

**(b)** $k = 10$ $R_{\text{acc},1} = 0.8334$ $R_{\text{acc},2} = 0.8335$

**(c)** $k = 20$ $R_{\text{acc},1} = 0.8869$ $R_{\text{acc},2} = 0.8870$

**(d)** $k = 50$ $R_{\text{acc},1} = 0.9370$ $R_{\text{acc},2} = 0.9370$

**(e)** $k = 100$ $R_{\text{acc},1} = 0.9602$ $R_{\text{acc},2} = 0.9604$

**(f)** $k = 150$ $R_{\text{acc},1} = 0.9704$ $R_{\text{acc},2} = 0.9713$

**Figure 1:** The SVD reconstruction of Des-SVD on *PeppersRGB* with different $k$ ($R_{\text{acc},1}$ is the SVD accuracy of Des-SVD and $R_{\text{acc},2}$ is that of the standard Jac-SVD).

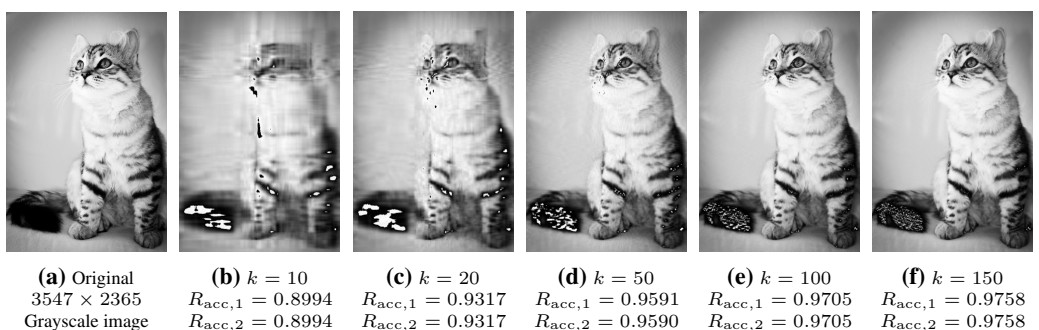

**(a)** Original $3547 \times 2365$ Grayscale image

**(b)** $k = 10$ $R_{\text{acc},1} = 0.8994$ $R_{\text{acc},2} = 0.8994$

**(c)** $k = 20$ $R_{\text{acc},1} = 0.9317$ $R_{\text{acc},2} = 0.9317$

**(d)** $k = 50$ $R_{\text{acc},1} = 0.9591$ $R_{\text{acc},2} = 0.9590$

**(e)** $k = 100$ $R_{\text{acc},1} = 0.9705$ $R_{\text{acc},2} = 0.9705$

**(f)** $k = 150$ $R_{\text{acc},1} = 0.9758$ $R_{\text{acc},2} = 0.9758$

**Figure 2:** The SVD reconstruction of Des-SVD on *Cat* with different $k$ ($R_{\text{acc},1}$ is the SVD accuracy of Des-SVD and $R_{\text{acc},2}$ is that of the standard Jac-SVD).

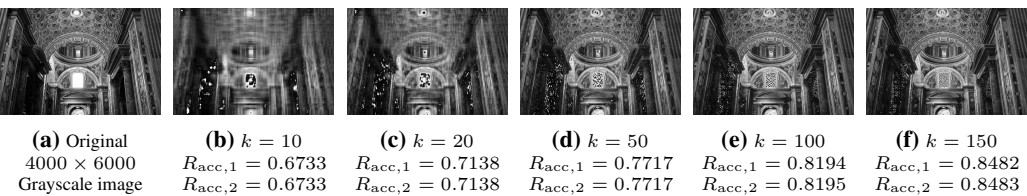

**(a)** Original $4000 \times 6000$ Grayscale image

**(b)** $k = 10$ $R_{\text{acc},1} = 0.6733$ $R_{\text{acc},2} = 0.6733$

**(c)** $k = 20$ $R_{\text{acc},1} = 0.7138$ $R_{\text{acc},2} = 0.7138$

**(d)** $k = 50$ $R_{\text{acc},1} = 0.7717$ $R_{\text{acc},2} = 0.7717$

**(e)** $k = 100$ $R_{\text{acc},1} = 0.8194$ $R_{\text{acc},2} = 0.8195$

**(f)** $k = 150$ $R_{\text{acc},1} = 0.8482$ $R_{\text{acc},2} = 0.8483$

**Figure 3:** The SVD reconstruction of Des-SVD on *Church* with different $k$ ($R_{\text{acc},1}$ is the SVD accuracy of Des-SVD and $R_{\text{acc},2}$ is that of the standard Jac-SVD).

In the above, we evaluate Des-SVD on both image data and random matrices. As discussed in Appendix F, Des-SVD remains stable and delivers accurate SVD results even in challenging scenarios, such as when the gap between two singular values is very small or when the condition number is exceptionally large.

**Table 3:** Performance Comparison of Different SVD Methods on Random Matrices.

| m, n | k | Lan-SVD | Jac-SVD | Des-SVD(Ours) |
|---|---|---|---|---|
| 500, 250 | 20 | $0.09 \pm 0.00s$ | $0.18 \pm 0.02s$ | $\mathbf{0.06 \pm 0.01s}$ |
| 750, 500 | 20 | $0.10 \pm 0.00s$ | $0.18 \pm 0.01s$ | $\mathbf{0.07 \pm 0.00s}$ |
| 1000,1000 | 50 | $\mathbf{0.14 \pm 0.02s}$ | $1.32 \pm 0.03s$ | $0.17 \pm 0.00s$ |
| 1500,2000 | 50 | $\mathbf{0.14 \pm 0.01s}$ | $1.37 \pm 0.04s$ | $0.44 \pm 0.00s$ |
| 3000,3000 | 100 | $\mathbf{0.32 \pm 0.02s}$ | $8.21 \pm 0.32s$ | $2.25 \pm 0.02s$ |
| 10000,10000 | 50 | $\mathbf{1.81 \pm 0.01s}$ | $13.59 \pm 0.01s$ | $1.97 \pm 0.01s$ |
| 10000,10000 | 100 | $\mathbf{2.84 \pm 0.01s}$ | $14.60 \pm 0.03s$ | $3.11 \pm 0.01s$ |

## 5.4 THE PARALLELIZATION PERFORMANCE OF DES-SVD

To further evaluate Des-SVD's parallelization performance,we present a time breakdown of Des-SVD, demonstrating that the process of estimating singular values and constructing the compatible matrix $D$ accounts for only a small fraction of the overall runtime, thereby highlighting the feasibility of parallelization. As shown in Table 4, we evaluate the performance using the image *Goldhill*[4] ($512 \times 512$, $k = 150$) and a matrix with power decay parameter $\alpha = 0.5$ ($100 \times 100$, $k = 100$).

**Table 4:** Time Cost Breakdown of Des-SVD
(The slowest stage is **bolded**, and the second slowest stage is *italicized*.)

| Time Stage | Hill.png | Matrix with power decay |
|---|---|---|
| Randomized subspace iteration | 0.1370s | 0.0087s |
| Rayleigh quotient iterations | 0.0096s | 0.0029s |
| Construction of compatible matrix | 0.0069s | 0.0024s |
| Initialization of shared memory | **0.3663s** | *0.0651s* |
| Newton method | 0.2207s | 0.0516s |
| Communication in parallel execution | *0.2384s* | **0.1234s** |
| **Total** | 0.9789s | 0.2541s |

Moreover, we compare the sequential and parallel implementations on *Baboon*[4] ($256 \times 256$) to demonstrate the speedup from our parallelization. As shown in Table 5, the speedup is modest for small $k$ but becomes considerable as $k$ increases. This is due to the fixed overhead from operations like shared memory preparation, making parallelization more beneficial for larger-scale computations.

**Table 5:** Parallel Performance of Des-SVD on Baboon.png

| $k$ | Sequential (s) | Parallel (s) | Speedup |
|---|---|---|---|
| 10 | 0.221 | 0.141 | 1.6 |
| 50 | 21.247 | 0.505 | 42.1 |
| 100 | 135.072 | 1.209 | 111.7 |

In addition to parallelization performance, we systematically evaluated the robustness and convergence of Des-SVD, as detailed in Appendix G. Results show that Des-SVD outperforms Jac-SVD in runtime while matching Lan-SVD, making it the first practical descent-based SVD algorithm.

---

[4]https://www.eecs.qmul.ac.uk/~phao/IP/Images/

## 6 CONCLUSIONS

By leveraging the primal–dual relationship between SVD and a least squares problem, we addressed a key challenge: among multiple minima arising from non-convexity, only one corresponds to the true SVD. Analyzing this, we found that the descent method could converge to the target solution by normalizing KKT solutions. Based on this, we developed Des-SVD, an efficient descent-based algorithm for SVD. Our experiments confirm that Des-SVD achieves performance comparable to matrix-based methods, with supplementary results in Appendix F - G.

While matrix-based methods remain mainstream, their limitations—especially in parallelization and distributed learning—highlight the need for alternatives. Descent methods show promise, but existing ones are too slow for practical use. Our Des-SVD is significantly more efficient than the Riemannian gradient method, offering a practical alternative. We hope this work paves the way for scalable descent-based methods for large-scale SVD in modern machine learning.

Furthermore, as stated in Theorem 3.2, a key condition for Des-SVD to obtain the true singular vectors is that the singular value estimation satisfies the threshold $T_{err}$. When this condition is met, the convergence follows the standard behavior of the Newton method. Otherwise, the objective value may become negative, indicating a failure of the decomposition. An interesting direction for future research is to investigate how to adapt or modify the singular value estimation—potentially improving the robustness of Des-SVD and enabling stability even when the estimation error exceeds the current threshold.

## 7 RELATED WORK

**Matrix-based SVD methods.** Currently, the dominant algorithms for solving SVD are matrix-based, mainly Jacobi's algorithm (Jacobi, 1846; Demmel & Veselic, 1992; Gao et al., 2025) and the Lanczos algorithm (Cullum et al., 1983; Cullum & Willoughby, 2006; Golub et al., 1981; Feng et al., 2018), along with several others (Nakatsukasa & Higham, 2013; Wang et al., 2021; Pialot et al., 2023). Efforts to accelerate these methods have largely focused on matrix approximation or low-level code optimization.

**Descent Methods for SVD.** Iterative descent techniques, including gradient descent (Jain et al., 2018), Newton's method (Chen et al., 2020; Polyak, 2007), and momentum methods (Liu et al., 2020; Qian, 1999), have become standard for large-scale problems. Riemannian gradient descent on the Stiefel manifold was introduced for SVD in 2013 (Sato & Iwai, 2013) and refined in recent works (Sato, 2014; Huang et al., 2025). However, it is slower than matrix-based methods due to the need for projection onto the manifold.

**Distributed SVD.** For distributed data, considerable efforts have been made to extend matrix-based methods (Hartebrodt et al., 2021; Chai et al., 2022; Blatt et al., 2020; Li et al., 2021). Nevertheless, most of these approaches still rely on collecting data at a central server for computation, which poses potential security risks, as highlighted by (Chai et al., 2024). Although Chai et al. (2024) further proposes a decentralized SVD method to improve security, the approach continues to incur high communication costs and cannot fully eliminate the need for data gathering and synchronization.

## ETHICS STATEMENT

This work focuses on developing and analyzing a novel descent-based method for singular value decomposition (Des-SVD). Our study is purely theoretical and experimental on synthetic and standard benchmark datasets, and does not involve human subjects, sensitive personal data, or applications with direct societal risks. We follow best practices to ensure reproducibility, and all code and experimental settings are made publicly available. We do not foresee any ethical concerns regarding the methodology or its applications within the scope of this work.

## REPRODUCTIVITY STATEMENT

To ensure reproducibility, we release the source code using the URL in the abstract. The README provides instructions for reproducing our results and implementing Des-SVD on arbitrary matrices. Theoretical foundations are discussed in Sections 2 and 3, with supplementary proofs in Appendices A and B.

## LLM USAGE

Large Language Models (LLMs) were used solely as writing assistants for improving the grammar, style, and clarity of the manuscript. They were not involved in the research ideation, design, theoretical development, implementation, or analysis. The authors take full responsibility for the content of this paper.

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

## TECHNICAL APPENDICES AND SUPPLEMENTARY MATERIAL

## A   THE PROOF OF ZERO-VALUE OF THE TARGET FUNCTION

From the KKT condition eq. (7), we can indicate by rearranging the terms that:

$$
\begin{cases}
\lambda \alpha_i = \sum_j \beta_j \psi(\boldsymbol{y}_j)^\top \varphi(\boldsymbol{x}_i), \forall i = 1, \ldots, n, \\
\lambda \beta_j = \sum_i \alpha_i \varphi(\boldsymbol{x}_i)^\top \psi(\boldsymbol{y}_j), \forall j = 1, \ldots, m,
\end{cases}
\tag{21}
$$

where $\lambda$ is the correct singular value and we have $\gamma = 1/\lambda$.

Substitute eq. (21) and eq. (7) into the objective function $\mathcal{J}$, then we have:

$$
\begin{aligned}
\mathcal{J} &= -\boldsymbol{w}^\top \boldsymbol{v} + \frac{1}{2}\gamma \sum_{i=1}^n e_i^2 + \frac{1}{2}\gamma \sum_{j=1}^m r_j^2 \\
&= -\sum_j \beta_j \psi(\boldsymbol{y}_j)^\top \sum_i \alpha_i \varphi(\boldsymbol{x}_i) + \frac{1}{2}\gamma \sum_{i=1}^n (\frac{\alpha_i}{\gamma})^2 + \frac{1}{2}\gamma \sum_{j=1}^m (\frac{\beta_j}{\gamma})^2 \\
&= \sum_i \alpha_i (-\sum_j \beta_j \psi(\boldsymbol{y}_j)^\top \varphi(\boldsymbol{x}_i)) + \frac{1}{2}\lambda \sum_{i=1}^n \alpha_i^2 + \frac{1}{2}\sum_{j=1}^m \lambda \beta_j^2 \\
&= \sum_{i=1}^n -\lambda \alpha_i^2 + \frac{1}{2}\lambda \sum_{i=1}^n \alpha_i^2 + \frac{1}{2}\sum_{j=1}^m \lambda \beta_j^2 \\
&= -\frac{1}{2}\lambda \boldsymbol{\alpha}^\top \boldsymbol{\alpha} + \frac{1}{2}\lambda \boldsymbol{\beta}^\top \boldsymbol{\beta}
\end{aligned}
\tag{22}
$$

If $\boldsymbol{\alpha} \in \mathbb{R}^n$ and $\boldsymbol{\beta} \in \mathbb{R}^m$ are the singlar vectors of the singular value $\lambda$ , we can obtain from the properties that $\boldsymbol{\alpha}^\top \boldsymbol{\alpha} = \boldsymbol{\beta}^\top \boldsymbol{\beta} = 1$, so the target function $\mathcal{J}$ will always be zero.

## B   THE PROOF THE THEOREMS IN SECTION 3

In this section, we will prove all the theorems mentioned in Section 3 respectively.

### B.1   THE PROOF OF THEOREM 3.1

The rank property of the KKT matrix $K$ given different $s$ is proved as follows:

We first define two matrices:

$$
\boldsymbol{K}_{\mathbf{up}} = \begin{bmatrix} \boldsymbol{H} & \boldsymbol{C}^\top \end{bmatrix} = \begin{bmatrix} \mathbf{0} & -\boldsymbol{I}_w & \mathbf{0} & \mathbf{0} & \boldsymbol{\Phi}^\top & \mathbf{0} \\ -\boldsymbol{I}_v & \mathbf{0} & \mathbf{0} & \mathbf{0} & \mathbf{0} & \boldsymbol{\Psi}^\top \\ \mathbf{0} & & \gamma \boldsymbol{I}_{e,r} & & -\boldsymbol{I}_{e,r} & \end{bmatrix} \in \mathbb{R}^{(3n+m)\times(4n+2m)},
\tag{23}
$$

$$
\boldsymbol{K}_{\mathbf{down}} = [\boldsymbol{C} \quad \mathbf{0}] \in \mathbb{R}^{(m+n)\times(4n+2m)},
\tag{24}
$$

Next, we eliminate $\boldsymbol{\Phi}^\top$ and $\boldsymbol{\Psi}$ by applying a row transformation with $-\boldsymbol{I}_{e,r}$, yielding the following:

$$
\boldsymbol{K}'_{\mathbf{up}} = \begin{bmatrix} \mathbf{0} & -\boldsymbol{I}_w & \gamma \boldsymbol{\Phi}^\top & \mathbf{0} & \mathbf{0} & \mathbf{0} \\ -\boldsymbol{I}_v & \mathbf{0} & \mathbf{0} & \gamma \boldsymbol{\Psi}^\top & \mathbf{0} & \mathbf{0} \\ \mathbf{0} & & \gamma \boldsymbol{I}_{e,r} & & -\boldsymbol{I}_{e,r} & \end{bmatrix}.
\tag{25}
$$

Define the matrix $\boldsymbol{B}$ as

$$\boldsymbol{B} = \begin{bmatrix} \boldsymbol{0} & -\boldsymbol{I_w} & \gamma \boldsymbol{\Phi}^\top & \boldsymbol{0} \\ -\boldsymbol{I_v} & \boldsymbol{0} & \boldsymbol{0} & \gamma \boldsymbol{\Psi}^\top \end{bmatrix} \in \mathbb{R}^{(m+n)\times(3n+m)}.$$

Then, the following equation constraints are obtained:

$$\boldsymbol{Bx} = \boldsymbol{0} \Leftrightarrow \begin{bmatrix} \boldsymbol{0} & -\boldsymbol{I_w} & \gamma \boldsymbol{\Phi}^\top & \boldsymbol{0} \\ -\boldsymbol{I_v} & \boldsymbol{0} & \boldsymbol{0} & \gamma \boldsymbol{\Psi}^\top \end{bmatrix} \begin{bmatrix} \boldsymbol{w} \\ \boldsymbol{v} \\ \boldsymbol{e} \\ \boldsymbol{r} \end{bmatrix} = \boldsymbol{0} \Leftrightarrow \begin{cases} \boldsymbol{v} = \sum_i \gamma e_i \varphi(\boldsymbol{x}_i), \forall i, \\ \boldsymbol{w} = \sum_j \gamma r_j \psi(\boldsymbol{y}_j), \forall j, \end{cases} \tag{26}$$

which is a different but equivalent form of the KKT condition. We define $\boldsymbol{K}'$ as

$$\boldsymbol{K}' = \begin{bmatrix} \boldsymbol{K}'_{\mathbf{up}} \\ \boldsymbol{K}_{\mathbf{down}} \end{bmatrix} \in \mathbb{R}^{(m+n)\times(4n+2m)}. \tag{27}$$

Since elementary row transformations do not affect the rank of a matrix, we have $\mathrm{rank}(\boldsymbol{K_{up}}) = \mathrm{rank}(\boldsymbol{K}'_{\mathbf{up}})$, i.e., $\mathrm{rank}(\boldsymbol{K}) = \mathrm{rank}(\boldsymbol{K}')$. Furthermore, if $\gamma$ is the correct value, eq. (14) leads to eq. (26) due to the KKT condition eq. (7). Therefore, we conclude that there exists at least one non-zero solution to the transformed KKT function by setting $\boldsymbol{v} = \boldsymbol{0}$ and $\boldsymbol{x} = \boldsymbol{x}^*$, where $\boldsymbol{x}^*$ is one of the non-zero KKT solutions:

$$\boldsymbol{K}' \begin{bmatrix} \boldsymbol{x}^* \\ \boldsymbol{0} \end{bmatrix} = \boldsymbol{0}. \tag{28}$$

In this case, $\boldsymbol{K}'$ is not of full rank, and neither is $\boldsymbol{K}$. Conversely, if $\gamma$ is incorrect, $\boldsymbol{K}$ becomes full rank because there is no non-zero solution that satisfies both $\boldsymbol{K}'_{\mathbf{up}}$ and $\boldsymbol{K}_{\mathbf{down}}$. This constraint restricts $\Delta\boldsymbol{x} = -\boldsymbol{x}$ to be the only solution to eq. (16).

## B.2 The proof of Theorem 3.2

The complete proof of the error threshold $T_{\mathrm{err}}$ is shown as follows:

We will prove Theorem 3.2 using proof by contradiction. For the true singular value $s$, it can be learnt from eq. (26) that :

$$\begin{cases} \boldsymbol{v} = \sum_i \frac{1}{s} e_i \varphi(\boldsymbol{x}_i), \forall i, \\ \boldsymbol{w} = \sum_j \frac{1}{s} r_j \psi(\boldsymbol{y}_j), \forall j. \end{cases} \tag{29}$$

Then, suppose there exists an estimate $\gamma' = \frac{1}{s} + \Delta\gamma'$ that satisfies eq. (19) but violates eq. (20). In other words, it must satisfy:

$$\|\Delta\gamma'\| < \min \left\{ \frac{\varepsilon_1}{(\sum_i \|\boldsymbol{D}^\top \boldsymbol{x}_i\|^2)^{\frac{1}{2}}}, \frac{\varepsilon_2}{(\sum_j \|\boldsymbol{y}_j\|^2)^{\frac{1}{2}}} \right\}, \tag{30}$$

and it should deviate from eq. (20):

$$\begin{cases} \|\boldsymbol{v} - \sum_i \gamma' e_i \varphi(\boldsymbol{x}_i)\| & \geq \varepsilon_1 \\ \|\boldsymbol{w} - \sum_j \gamma' r_j \psi(\boldsymbol{y}_j)\| & \geq \varepsilon_2. \end{cases} \tag{31}$$

Since $\boldsymbol{\Phi}$ and $\boldsymbol{\Psi}$ are simply the feature mappings of the row and column vectors eq. (3) of the target matrix $\boldsymbol{A}$, we can rewrite eq. (31) as:

$$\begin{cases} \|\boldsymbol{v} - \sum_i \gamma' e_i \boldsymbol{D}^\top \boldsymbol{x}_i\| & \geq \varepsilon_1 \\ \|\boldsymbol{w} - \sum_j \gamma' r_j \boldsymbol{y}_j\| & \geq \varepsilon_2. \end{cases} \tag{32}$$

Then simplify eq. (32) with the numerical eliminations of $1/s$:

$$\begin{cases} \|\Delta\gamma' \sum_i e_i (\boldsymbol{D}^\top \boldsymbol{x}_i)\| & \geq \varepsilon_1 \\ \|\Delta\gamma' \sum_j v_j \boldsymbol{y}_j\| & \geq \varepsilon_2. \end{cases} \tag{33}$$

Because of the normalization on $\boldsymbol{e} = [e_1, ..., e_n]$ and $\boldsymbol{v} = [v_1, ..., v_m]$ (see Section 3.1), the following equations hold:

$$\begin{cases} \sum_i e_i^2 = 1 \\ \sum_j v_j^2 = 1. \end{cases} \tag{34}$$

Combined with the Cauchy-Schwarz inequality, we have:

$$\left\| \sum_i e_i \boldsymbol{D}^\top \boldsymbol{x}_i \right\| \leq \left( \sum_i e_i^2 \right)^{\frac{1}{2}} \left( \sum_i \|\boldsymbol{D}^\top \boldsymbol{x}_i\|^2 \right)^{\frac{1}{2}} = \left( \sum_i \|\boldsymbol{D}^\top \boldsymbol{x}_i\|^2 \right)^{\frac{1}{2}}. \tag{35}$$

By applying a similar derivation to $\boldsymbol{v}$, we can now conclude that:

$$\| \sum_i e_i \boldsymbol{D}^\top \boldsymbol{x}_i \| \leq ( \sum_i \|\boldsymbol{D}^\top \boldsymbol{x}_i\|^2 )^{\frac{1}{2}} \quad \text{or} \quad \| \sum_j v_j \boldsymbol{y}_j \| \leq ( \sum_j \|\boldsymbol{y}_j\|^2 )^{\frac{1}{2}}. \tag{36}$$

If we define the former equation in eq. (36) as (a) and the later one as (b), we can conclude that $\|\Delta\gamma'\|$ should be larger than $\frac{\varepsilon_1}{(\sum_i |\boldsymbol{D}^\top \boldsymbol{x}_i|^2)^{\frac{1}{2}}}$ if (a) satisfies or larger than $\frac{\varepsilon_2}{(\sum_j |\boldsymbol{y}_j|^2)^{\frac{1}{2}}}$ if (b) satisfies.

Therefore, we derive the lower bound of $\|\Delta\gamma'\|$:

$$\|\Delta\gamma'\| \geq \min \left\{ \frac{\varepsilon_1}{(\sum_i \|\boldsymbol{D}^\top \boldsymbol{x}_i\|^2)^{\frac{1}{2}}}, \frac{\varepsilon_2}{(\sum_j \|\boldsymbol{y}_j\|^2)^{\frac{1}{2}}} \right\}, \tag{37}$$

which is contradictory to eq. (30). Thus, we have established the validity of the proposition on $T_{\text{err}}$.

## C    The refined version of the SVD algorithm (Des-SVD)

In this section, we present the refined Descent SVD method (Des-SVD). The algorithm integrates parallelization and randomized sampling, allowing each singular value to be computed independently and efficiently. It first estimates the leading singular values, then solves the associated KKT systems for the singular vectors, and finally assembles the complete SVD solution. The detailed procedure is summarized in Algorithm 2.

## D    The Methods for Fast Singular Value Approximation

We adopt the Rayleigh Quotient Iteration method in the paper to fast estimate singular values, and its concrete realization is demonstrated in Algorithm 3.

---

**Algorithm 2** The refined Descent SVD Method (Des-SVD).

---

**Input:** $A \in \mathbb{R}^{n \times m}$ : The target matrix ; $k$: the number of singular values to be calculated; $n_{\max}$: the max iterations for Newton's method; $\varepsilon$: the threshold for convergence ; `parallel` : whether to adapt parallelization ; `randomized`: whether to use random sampling .

**Output:** $[\boldsymbol{\alpha}]$, $\boldsymbol{S}$, $[\boldsymbol{\beta}]$: the SVD solution of $A$

1: **if** `randomized` **then**
2:     Calculate the orthonormal matrix $\boldsymbol{Q} = $ `Randomized-Subspace-Iteration`$(\boldsymbol{A})$.
3:     Get the low-rank approximated matrix $\boldsymbol{A}' = \boldsymbol{Q}^* \boldsymbol{A}$.
4: **else**
5:     Set $\boldsymbol{A}' = \boldsymbol{A}$.
6: **end if**
7: Initialize the dual variables $[\boldsymbol{\alpha}] \in \mathbb{R}^{k \times n}$ and $[\boldsymbol{\beta}] \in \mathbb{R}^{k \times m}$.
8: Estimate the first $k$ singular values and store them in $\boldsymbol{S} = $ `Rayleigh-Quotient`$(\boldsymbol{A}', k)$.
9: Construct the coefficient matrix $\boldsymbol{C} \in \mathbb{R}^{(m+n) \times (3n+m)}$ such that $\boldsymbol{Cx} = \boldsymbol{0}$.
10: **if** `parallel` **then**
11:     Create $k$ processes for each singular value $s \in \boldsymbol{S}$.
12:     **for** each singular value $s_i$ in its **independent** process $p_i$ **do**
13:         Execute Algorithm 1 with **Input**: $\{\boldsymbol{A}', \boldsymbol{C}, 1/s_i, n_{\max}, \varepsilon\}$ and **Output**: $\{\boldsymbol{\alpha}'_i, \boldsymbol{\beta}'_i\}$.
14:     **end for**
15:     Gather all the results together in order, i.e., $[\boldsymbol{\alpha}'] = [\boldsymbol{\alpha}'_1, ..., \boldsymbol{\alpha}'_k]^\top$ and $[\boldsymbol{\beta}'] = [\boldsymbol{\beta}'_1, ..., \boldsymbol{\beta}'_k]^\top$.
16: **else**
17:     **for** each singular value $s_i \in S$ **do**
18:         Execute Algorithm 1 with **Input**: $\{\boldsymbol{A}', \boldsymbol{C}, 1/s_i, n_{\max}, \varepsilon\}$ and **Output**: $\{\boldsymbol{\alpha}'_i, \boldsymbol{\beta}'_i\}$.
19:         Update $[\boldsymbol{\alpha}']$.`iloc`$(i, :) = \boldsymbol{\alpha}'_i$ and $[\boldsymbol{\beta}']$.`iloc`$(i, :) = \boldsymbol{\beta}'_i$.
20:     **end for**
21: **end if**
22: **if** `randomized` **then**
23:     Update $[\boldsymbol{\alpha}] = \boldsymbol{Q}[\boldsymbol{\alpha}']$ and remain $[\boldsymbol{\beta}] = [\boldsymbol{\beta}']$.
24: **else**
25:     Remain $[\boldsymbol{\alpha}] = [\boldsymbol{\alpha}']$ and $[\boldsymbol{\beta}] = [\boldsymbol{\beta}']$.
26: **end if**
27: **Return** $[\boldsymbol{\alpha}]$, $\boldsymbol{S}$, $[\boldsymbol{\beta}]$

---

**Algorithm 3** Estimate the top-k singular values using Rayleigh quotient iteration

---

**Input:** $A \in \mathbb{R}^{n \times m}$, number of singular values $k$, max iteration number $n_{\text{iter}}$, convergence threshold $\varepsilon_{\text{rayleigh}}$.

**Output:** Top-k singular values of $A$

1: Initialize random matrix $\boldsymbol{V} \in \mathbb{R}^{m \times k}$ such that $\boldsymbol{V}$ is orthogonal.
2: **for** $i = 1$ **to** $n_{\text{iter}}$ **do**
3:     $\boldsymbol{Z} = \boldsymbol{A}^T \boldsymbol{A} \boldsymbol{V}$
4:     $\boldsymbol{V}_{\text{new}}, \_ = $ `QR-Factorization`$(\boldsymbol{Z})$
5:     **if** $\|\boldsymbol{V}_{\text{new}} - \boldsymbol{V}\| < \varepsilon_{\text{rayleigh}}$ **then**
6:         **break**
7:     **end if**
         $\boldsymbol{V} = \boldsymbol{V}_{\text{new}}$
8: **end for**
9: Compute singular values by taking the $L_2$ norm of each column of the matrix product $\boldsymbol{C} = \boldsymbol{AV}$.
10: Sort singular values in descending order
11: **Return** sorted singular values

---

# E HYPERPARAMETER SELECTION AND ABLATIONS

## E.1 HYPERPARAMETER SETTINGS FOR COMPARATIVE METHODS

For the three comparison methods(Rie-SVD, Lan-SVD, and Jac-SVD), we employ standard hyperparameter settings. For Rie-SVD, we adopt the standard configuration from (Sato & Iwai, 2013) with $\alpha_{\min} = 1 \times 10^{-6}$,

$\alpha_{\max} = 1.0$, max-iter $= 50000$, $\beta_m = 0.5$, $\epsilon = 0.5$, and $\epsilon_f = 1 \times 10^{-10}$. For Jac-SVD, we use the default hyperparameters $\varepsilon = 1 \times 10^{-6}$, $n_{\max} = 100$, and `randomized` = True, while Lan-SVD is configured with $\varepsilon = 1 \times 10^{-6}$, $n_{\max} = 100$, and `randomized` = True.

## E.2 THE ABLATION STUDY OF THE NEWTON METHOD IN DES-SVD

For the ablation study of the parameters in Des-SVD, particularly in the Newton method, we conduct the following experiment. We select a $100 \times 100$ matrix with exponential decay $\beta = 0.5$. Several groups of common parameters are chosen, i.e., $n_{\max} \in \{1, 3, 5, 10\}$ and $(\alpha, \beta) \in \{(0.1, 0.8), (0.01, 0.99), (0.2, 0.7), (0.5, 0.5)\}$. The ablation experiment results are shown in Table 6.

**Table 6:** Ablation Study of Newton Method Parameters

| $n_{\max}$ | $\alpha$ | $\beta$ | $R_{\mathrm{acc}}$ | Time (sec) |
|---|---|---|---|---|
| 1 | 0.1 | 0.8 | 0.9952 | 0.1137 |
| 3 | 0.1 | 0.8 | 0.9952 | 0.1507 |
| 5 | 0.1 | 0.8 | 0.9952 | 0.1782 |
| 10 | 0.1 | 0.8 | 0.9952 | 0.2579 |
| 1 | 0.01 | 0.99 | 0.9952 | 0.4233 |
| 3 | 0.01 | 0.99 | 0.9952 | 1.0242 |
| 5 | 0.01 | 0.99 | 0.9952 | 1.6732 |
| 10 | 0.01 | 0.99 | 0.9952 | 3.2992 |
| 1 | 0.2 | 0.7 | **0.9952** | **0.0871** |
| 3 | 0.2 | 0.7 | 0.9952 | 0.1256 |
| 5 | 0.2 | 0.7 | 0.9952 | 0.2340 |
| 10 | 0.2 | 0.7 | 0.9952 | 0.2878 |
| 1 | 0.5 | 0.5 | 0.9952 | 0.1182 |
| 3 | 0.5 | 0.5 | 0.9952 | 0.1608 |
| 5 | 0.5 | 0.5 | 0.9952 | 0.1769 |
| 10 | 0.5 | 0.5 | 0.9952 | 0.2543 |

Overall, the performance is not highly sensitive to the backtracking parameters, although there are some small differences. We recommend using $\alpha = 0.2$ and $\beta = 0.7$, which are the values we use in all of our experiments. Additionally, we set $n_{\max} = 3$. While we cannot theoretically claim that 3 iterations guarantee convergence, the accuracy achieved with this setting is sufficient to provide an accurate SVD.

## E.3 HYPERPARAMETER SELECTION IN DES-SVD

We specify the hyperparameters configured for each computational stage of Des-SVD.

- **Randomized subspace iteration**: Following Algorithm 4.4 in Halko et al. (2011), we compute the orthonormal matrix $Q$ with the number of power iterations set to $q = 5$.

- **Rayleigh quotient iteration**: We configure $n_{\mathrm{iter}} = 3$ and the tolerance $\varepsilon_{\mathrm{rayleigh}} = 1 \times 10^{-6}$ (see Appendix G for detailed analysis).

- **Newton method**: Based on the ablation study in Appendix E.2, we employ $n_{\max} = 3$, $\alpha = 0.7$, and $\beta = 0.2$ in practice. Given the satisfactory convergence behavior, we set $\varepsilon = 1 \times 10^{-6}$.

## F  SVD ON MORE SPECIAL CASES

To assess the performance of Des-SVD in a more general scenario, we perform experiments using two different singular value decay models: power-law decay ($\sigma_k \sim k^{-\alpha}$) and exponential decay ($\sigma_k \sim Ce^{-\beta k}$), with an initial singular value of $\sigma_1 = 10^4$ and matrix dimensions of $m = n = 100$. Since the accuracy across all methods is nearly identical, we focus primarily on the computational efficiency, presenting only the time performance results in Tables 7 and 8. As observed in Table 8, even when the condition number is large, our method demonstrates performance on par with Lan-SVD and outperforms Jac-SVD, highlighting its stability and resilience under difficult conditions.

**Table 7:** Performance Comparison of Different SVD Methods on Matrices Following Power-law Decay .

| $\alpha$ | Lan-SVD | Jac-SVD | Des-SVD(Ours) | Condition Number |
|---|---|---|---|---|
| 0.5 | **0.16 ± 0.00s** | 3.51 ± 0.08s | 0.17 ± 0.00s | 10.0 |
| 1.0 | **0.14 ± 0.02s** | 2.56 ± 0.04s | 0.19 ± 0.00s | 100.0 |
| 1.2 | 0.24 ± 0.03s | 2.30 ± 0.13s | **0.16 ± 0.01s** | 252.1 |
| 1.5 | **0.16 ± 0.03s** | 2.29 ± 0.01s | 0.21 ± 0.00s | 1000.0 |

**Table 8:** Performance Comparison of Different SVD Methods on Matrices Following Exponential Decay.

| $\beta$ | Lan-SVD | Jac-SVD | Des-SVD(Ours) | Condition Number |
|---|---|---|---|---|
| 0.2 | **0.14 ± 0.00s** | 2.39 ± 0.13s | 0.21 ± 0.00s | $5 \times 10^9$ |
| 0.5 | **0.14 ± 0.00s** | 5.41 ± 0.07s | 0.20 ± 0.00s | $8 \times 10^9$ |
| 0.8 | **0.13 ± 0.00s** | 6.89 ± 0.47s | 0.16 ± 0.00s | $8.8 \times 10^9$ |
| 1.0 | **0.15 ± 0.01s** | 6.78 ± 0.24s | 0.16 ± 0.00s | $2.3 \times 10^{10}$ |

Furthermore, to test the orthogonality of the singular vector matrices corresponding to nearly identical singular values, we design additional experiments. For a fixed matrix size of $(m, n) = (100, 100)$, we select the top-$k$ singular values and set them as follows:

$$S[: k] = \text{Descending\_Sorted}(s_{\max} \cdot (1 + \epsilon \cdot t)), \tag{38}$$

where $s_{\max}$ is the largest singular value, $\epsilon$ controls the level of similarity, and $t \sim N(0, 1)$ is drawn from a normal distribution. To further test the orthogonality, we calculate the mean deviation from orthogonality for both $U$ and $V$, representing the left and right singular vector matrices, respectively. Let $X \in \mathbb{R}^{m \times d}$, and the mean deviation from orthogonality is defined as follows:

$$\text{MDO}(X) = \frac{1}{d} \left( \|X^T X - I\|_F \right), \tag{39}$$

where $d$ is the number of columns in $X$, $m$ is the number of rows, and $I$ is the identity matrix of size $d \times d$. This metric measures the degree to which $X$ deviates from being orthogonal.

**Table 9:** Comparison of U and V Orthogonality Errors and Accuracy for Different $\epsilon$ and $k$ Values.

| $\epsilon$ | $k = 5$ | | | $k = 10$ | | | $k = 20$ | | |
|---|---|---|---|---|---|---|---|---|---|
| | $R_{\text{acc}} \uparrow$ | MDO($U$)↓ | MDO($V$)↓ | $R_{\text{acc}} \uparrow$ | MDO($U$)↓ | MDO($V$)↓ | $R_{\text{acc}} \uparrow$ | MDO($U$)↓ | MDO($V$)↓ |
| 0.1 | 99.42% | 4.093E-04 | 4.097E-04 | 97.29% | 5.732E-04 | 5.734E-04 | 98.76% | 3.079E-04 | 3.081E-04 |
| 0.01 | 99.01% | 3.838E-04 | 3.844E-04 | 99.00% | 4.170E-04 | 4.172E-04 | 98.22% | 5.324E-04 | 5.324E-04 |
| 0.001 | 86.98% | 6.311E-04 | 6.312E-04 | 80.14% | 9.988E-04 | 9.984E-04 | 68.12% | 1.946E-03 | 1.947E-03 |

The experimental results for different $k$ and $\epsilon$ are shown in Table 9. As $\epsilon$ decreases (indicating higher similarity between singular values) and $k$ increases (introducing more similar singular values), reconstruction

accuracy decreases. However, the orthogonality of $U$ and $V$ remains well-preserved, demonstrating the method's stability and robustness in maintaining the orthogonality of the singular vectors. This underscores the effectiveness of our approach in preserving the decomposition structure. For $\epsilon < 1e - 3$, singular values are considered effectively identical, as their differences become negligible.

## G    ROBUSTNESS ANALYSIS AND CONVERGENCE GUARANTEES

Regarding convergence, when Des-SVD correctly solves the SVD, the convergence follows the standard Newton method. If it does not converge, the objective value rapidly becomes negative, which provides a clear signal to terminate the algorithm, as shown in Algorithm 1.

For robustness, several factors may be considered. We have evaluated the behavior of the method under different singular value decay rates and varying ranks, and we have also examined the case where two singular values are close to each other (as detailed in Appendix F).

In addition, a specific robustness issue in Des-SVD is the singular value estimation. Here, we first evaluate the performance of the Rayleigh method using different numbers of iterations.

In practice, rather than focusing on one singular value as theoretically analyzed in Section 3.3, we use $Err_{\text{avg}}(S_{\text{es}})$ to describe the average estimation error. Let $S$ denote the true singular value matrix and $S_{\text{es}}$ the estimated one. We define the average estimation error as:

$$Err_{\text{avg}}(S_{\text{es}}) = \frac{1}{k}\|S_{\text{es}} - S\|_F,$$

where $k$ is the number of singular values. Experiments in Table 10 and Table 11 show that the Rayleigh iteration method converges effectively, and we choose $n_{\text{iter}} = 3$ for all reported experiments. We also report the maximum and minimum values of estimation error across all singular values to demonstrate that the estimation error is well-balanced and has minimal impact on different singular values.

**Table 10:** Rayleigh Iteration Performance on Hill.png

| $n_{\text{iter}}$ | $Err_{\text{avg}}(S_{\text{rayleigh}})$ | $Err_{\text{max}}(S_{\text{rayleigh}})$ | $Err_{\text{min}}(S_{\text{rayleigh}})$ | Time (sec) |
|---|---|---|---|---|
| 1 | $3.5864 \times 10^{-7}$ | $1.6000 \times 10^{-5}$ | $< 1.0000 \times 10^{-7}$ | $3.7 \times 10^{-3}$ |
| 3 | $2.1186 \times 10^{-7}$ | $1.1000 \times 10^{-5}$ | $< 1.0000 \times 10^{-7}$ | $5.6 \times 10^{-3}$ |
| 10 | $1.8267 \times 10^{-7}$ | $1.0000 \times 10^{-5}$ | $< 1.0000 \times 10^{-7}$ | $1.2 \times 10^{-2}$ |

**Table 11:** Rayleigh Iteration Performance on Matrix with Exponential Decay

| $n_{\text{iter}}$ | $Err_{\text{avg}}(S_{\text{rayleigh}})$ | $Err_{\text{max}}(S_{\text{rayleigh}})$ | $Err_{\text{min}}(S_{\text{rayleigh}})$ | Time (sec) |
|---|---|---|---|---|
| 1 | $6.8593 \times 10^{-5}$ | $3.6620 \times 10^{-3}$ | $< 1.0000 \times 10^{-7}$ | $4.2 \times 10^{-3}$ |
| 3 | $6.6933 \times 10^{-5}$ | $3.1740 \times 10^{-3}$ | $< 1.0000 \times 10^{-7}$ | $5.5 \times 10^{-3}$ |
| 5 | $5.8387 \times 10^{-5}$ | $3.1740 \times 10^{-3}$ | $< 1.0000 \times 10^{-7}$ | $7.3 \times 10^{-3}$ |

We can observe that the average estimation error is approximately within $1 \times 10^{-4}$. Next, we evaluate the SVD performance based on $R_{\text{acc}}$ for different estimation accuracies at this error level. Here, the estimation error is artificially introduced by adding Gaussian noise to the estimated singular value. This is based on our observation that such noise has a uniform effect on singular values, regardless of their magnitude. Specifically, we define the singular value matrix with Gaussian noise as $S_{\text{noise}}(b) = S + bE$, where each component

$E_{ij} \sim N(0,1)$ represents Gaussian noise. The results in Table 12 show that our method exhibits robust performance against estimation error.

Table 12: The Performance of Singular Value Estimation under Different Noise Levels

| Data | m,n | k | b = 0 | b = $1 \times 10^{-5}$ | b = $1 \times 10^{-4}$ | b = $1 \times 10^{-3}$ |
|---|---|---|---|---|---|---|
| Baboon | 256,256 | 100 | 0.9072 | 0.9072 | 0.90710 | 0.9003 |
| Goldhill | 512,512 | 100 | 0.9612 | 0.9611 | 0.9578 | 0.9511 |
| Power decay $\alpha = 0.5$ | 100,100 | 100 | 0.9979 | 0.9978 | 0.9978 | 0.9965 |
| Exp. decay $\beta = 0.5$ | 1000,1000 | 250 | 0.9999 | 0.9999 | 0.9999 | 0.9998 |

