# OpenReview forum: "A Practical Descent Method for Singular Value Decomposition"
_ICLR.cc/2026/Conference — Submitted to ICLR 2026_

### Official Review · Reviewer_bebZ · 2025-10-31

**Soundness:** 3
**Presentation:** 3
**Contribution:** 3
**Rating:** 4
**Confidence:** 3

**Summary:**

The paper proposes a practical descent-based method for computing the Singular Value Decomposition (SVD) that avoids traditional matrix-centric pipelines and costly manifold projections used by Riemannian methods. Building on Suykens (2016), the authors formulate a primal least-squares problem whose dual corresponds to SVD and derive KKT conditions that bridge the two spaces. The key contributions are: 1) Showing that when the regularization $\gamma$ equals the reciprocal of a singular value $s$, any non-zero KKT solution can be normalized to yield the corresponding singular vectors, with orthogonality arising naturally. 2) Proving that if $\gamma$ is incorrect (i.e., not $1/s$), the KKT matrix is full-rank so the only feasible descent direction at initialization leads to a zero solution—highlighting the need for accurate or sufficiently accurate singular value estimates. 3) Establishing that inexact singular value estimates suffice under explicit reconstruction tolerances $\epsilon_1$, $\epsilon_2$, yielding bounds on allowable $\gamma$-error ($T_{\mathrm{err}}$). This enables the use of fast eigenvalue estimators (Rayleigh quotient iteration). 4) Designing Des-SVD: a Newton-based descent algorithm that (i) estimates singular values, (ii) solves the primal KKT system to obtain dual variables, and (iii) normalizes them to get singular vectors. The method supports parallelization (each singular value independently) and randomized subspace reduction, giving overall complexity comparable in order to Lanczos ($O(k^3)$ in the reduced $k$-space).

Empirically, Des-SVD is much faster than Riemannian gradient descent and competitive with matrix-based methods (Lanczos, randomized + Jacobi) on low-rank matrices, grayscale/RGB images, random large matrices, and challenging spectra (tight gaps, large condition numbers). The authors report stable orthogonality and reconstruction accuracy across settings.

**Strengths:**

Originality: 1) A clear primal–dual route to SVD that avoids manifold projections, built upon but extending Suykens (2016) with a practical path from primal KKT solutions to SVD factors. 2) Theoretical identification of when the KKT system admits non-zero solutions ($\gamma=1/s$) and when it collapses (incorrect $\gamma$), plus a tolerance-bound analysis that legitimizes inexact singular value estimation. This is a neat, useful bridge between optimization feasibility and spectral approximation. 3) The integration of randomized subspace methods and process-level parallelization into a descent framework is well motivated and practically relevant.

Quality: 1) Theoretical results are clear and aligned with algorithmic design: (i) non-zero KKT solutions imply orthogonal singular vectors after normalization; (ii) rank analysis of the KKT matrix; (iii) error tolerance $T_{\mathrm{err}}$ tied to reconstruction budgets. The proofs are outlined in the main text with appendices for details. 2) Algorithmic choices are coherent: Newton’s method for the small KKT systems after sampling; Rayleigh quotient iteration for singular value estimates; per-singular-value parallelization. 3) Empirical evaluation covers low-rank matrices, images, random large matrices, and special spectra, consistently comparing against representative baselines (Riemannian GD, Lanczos, randomized SVD + Jacobi).

Clarity: 1) The narrative clearly states the obstacle with non-convexity and how the proposed normalization and $\gamma$-selection resolve it. The role of $\gamma$ is emphasized repeatedly with proofs to support intuition. 2) Pseudocode for core routines (Algorithm 1–3) and method variants (refined, parallel, randomized) helps reproducibility.

Significance: 1) Descent-based SVD that is competitive with classical matrix methods—while being naturally parallel and sampling-friendly—addresses an important gap for large-scale, distributed, or privacy-conscious settings where matrix factorization pipelines are awkward. 2) The tolerance-based theory provides a practical recipe for integrating fast spectral estimation within a descent solver, which could influence future work in primal–dual spectral algorithms and distributed SVD/EVD.

**Weaknesses:**

Dependence on singular value estimation: While Theorem 3.2 justifies inexact $\gamma$ within $T_{\mathrm{err}}$, the practical tightness of $T_{\mathrm{err}}$ and its dependence on data norms $(\sum_i\|D^{T}x_i\|^2, \sum_{j}\|y_j\|^2)$ may be conservative. The paper does not quantify how often Rayleigh iteration meets this tolerance in practice, how many iterations it requires for various spectra, or how sensitive overall runtime is to this stage.

Role and construction of ${D}$ (feature mapping): The feature map uses a matrix ${D}$ satisfying ${ADA} = {A}$. The paper does not discuss concrete choices for $D$, their computational cost, or sensitivity. If $D\approx I$ is intended in many cases, clarify; if not, provide guidance on constructing $D$ efficiently and its effect on $\Phi, \Psi$ conditioning and $T_{\mathrm{err}}$.

Complexity and scaling details: The stated $O((6k)^3)$ complexity after reduction is appealing, but a more granular cost breakdown would help: cost of randomized subspace iteration, Rayleigh quotient iterations, number of Newton steps per singular vector, linear solver costs for the KKT system, and communication in parallel execution.

Numerical robustness and stopping criteria: The descent loop stops when $\|J\|^2<\epsilon$, but there is limited discussion on i) Conditioning of the KKT system and necessity of regularization or preconditioning; ii) Line search strategy specifics and safeguards against stagnation; iii) How normalization interacts with Newton steps across iterations (e.g., re-scaling $e$ and $v$ each iteration can alter curvature; is this accounted for in convergence guarantees?).

Comparisons and breadth: 1) The baseline “randomized + Jacobi” is reasonable, but state-of-the-art randomized SVD implementations often avoid full Jacobi at the end. Including a power-iteration or subspace-iteration-based RSVD with small dense SVD on the core would be informative. 2) GPU or multi-node settings are hinted as strengths; however, all experiments are CPU-only. A small-scale demonstration of parallel speedups or scalability with increasing number of cores/nodes would better support the parallelization claims.

Theoretical scope: 1) The non-convex landscape: the paper shows that correct $\gamma$ enables non-zero KKT solutions aligned with singular vectors, but it does not analyze basin of attraction, global convergence, or robustness to noise/perturbations beyond $T_{\mathrm{err}}$. A brief discussion of potential spurious stationary points and how the algorithm avoids them (beyond $\gamma$ correctness) would help.

**Questions:**

Singular value estimation and $T_{\mathrm{err}}$: 1.  How tight is $T_{\mathrm{err}}$ in practice? Can you report empirical $T_{\mathrm{err}}$ values alongside achieved $\gamma$-errors from Rayleigh iteration across datasets, and the number of iterations needed to meet $T_{\mathrm{err}}$? 2. Does the algorithm have an adaptive mechanism to refine $\gamma$ if the KKT system indicates near-full-rank behavior (e.g., small but nonzero residuals), instead of collapsing to zero on the first step? For instance, can you detect ill-conditioning of $\kappa$ and trigger another Rayleigh update?

Choice and construction of $D$: 1. What concrete $D$ do you use in experiments? Explain the construction and its cost. 2. How does $D$ affect $T_{\mathrm{err}}$ and the conditioning of $\Phi, \Psi$? Could $D$ be tuned to improve numerical stability or accelerate convergence?

Newton solver and numerical stability: 1. Which linear solver and preconditioning strategy do you use for the KKT system? Any regularization applied to handle near-singularity when $\gamma$ is close but not exact? 2. Can you provide ablations on the number of Newton iterations ($n_{\mathrm{max}}$), line search parameters, and their impact on both accuracy and runtime?

Extensions and limitations: 1. Could the framework extend to computing singular subspaces directly (block methods), potentially improving robustness for clustered singular values? 2. What are the failure modes you observed (e.g., near-identical singular values with noise), and can you propose diagnostics or automatic restarts?

Theoretical clarifications:  1. In Theorem 3.1, you show full-rank $K$ when $\gamma$ is incorrect. Is there a quantitative condition number characterization as $\gamma$ approaches $1/s$, to explain practical behavior before collapse? That could guide adaptive $\gamma$ updates. 2. In Theorem 3.2, the bound uses Cauchy–Schwarz over sums of features. Are there tighter, data-dependent bounds (e.g., using coherence or leverage scores) that give less conservative $T_{\mathrm{err}}$ in practice?

Reproducibility and code: Please clarify the exact randomized subspace iteration routine and parameters (power iters, oversampling $p$). Also include the specific Rayleigh quotient iteration settings (niter, tolerance), and the line search strategy details.

Overall suggestion: Add a section with (i) an adaptive $\gamma$-refinement loop informed by KKT residuals/conditioning, (ii) more thorough ablations on $D$, Newton iterations, and RSVD parameters, and (iii) scaling experiments that demonstrate parallel gains. These would substantially strengthen the practical case for Des-SVD.

---

> ### Author Response · Authors · 2025-11-20
> **Response to Reviewer bebZ (1/4)**
>
> We sincerely thank you for your constructive and valuable comments. Accordingly, we address each of your concerns in detail as follows.
>
> ---
>
> **Singular Value Estimation and Accuracy (Q1)**
>
> Thank you for your questions regarding singular value estimation and accuracy. We would like to address them in the following two points respectively.
>
> ### 1. For "the practical value of $T_{err}$ and how the Rayleigh iteration meets $T_{err}$"
>
> In practice, the value of $T_{err}$ is difficult to pre-define. Instead, we report the average estimation error across different Rayleigh iterations and design a new experiment to verify that within 3 iterations, the estimation accuracy reaches a satisfactory level, thus providing an empirically observed bound.
>
> In the following, we use $Err_{avg}(S_{es})$ to describe the average estimation error. Let $S$ denote the true singular value matrix and $S_{es}$ the estimated one. We define the average estimation error as:
>
> $$
> Err_{avg}(S_{es}) = \frac{1}{k} \| S_{es} - S \|_F,
> $$
>
> where $k$ is the number of singular values. Experiments in **Table 1** and **Table 2** show that the Rayleigh iteration method converges effectively, and we choose $n_{iter} = 3$ for all reported experiments. We also report the maximum and minimum values of estimation error across all singular values to demonstrate that the estimation error is well-balanced and has minimal impact on different singular values.
>
> **Table 1: Rayleigh Iteration Performance on Hill.png**
>
> **Data**: Hill.png, $k = 100$.
>
> | $n_{iter}$ | $Err_{avg}(S_{rayleigh})$ | $Err_{max}(S_{rayleigh})$ | $Err_{min}(S_{rayleigh})$ | Time (sec)           |
> | ---------- | ------------------------- | ------------------------- | ------------------------- | -------------------- |
> | 1  | $3.5864 \times 10^{-7}$  | $1.6000 \times 10^{-5}$ | $< 1.0000 \times 10^{-7}$ | $3.7 \times 10^{-3}$ |
> | 3  | $2.1186 \times 10^{-7}$ | $1.1000 \times 10^{-5}$ | $< 1.0000 \times 10^{-7}$ | $5.6 \times 10^{-3}$ |
> | 10 | $1.8267 \times 10^{-7}$ | $1.0000 \times 10^{-5}$ | $< 1.0000 \times 10^{-7}$ | $1.2 \times 10^{-2}$ |
>
> **Table 2: Rayleigh Iteration Performance on Matrix with Exponential Decay**
>
> **Data**: Matrix(100 $\times$ 100) with power decay $\alpha = 0.5$, $k = 100$.
>
> | $n_{iter}$ | $Err_{avg}(S_{rayleigh})$ | $Err_{max}(S_{rayleigh})$ | $Err_{min}(S_{rayleigh})$ | Time (sec)           |
> | ---------- | ---------------------------- | ---------------------------- | ---------------------------- | -------------------- |
> | 1 | $6.8593 \times 10^{-5}$      | $3.6620 \times 10^{-3}$      | $< 1.0000 \times 10^{-7}$    | $4.2 \times 10^{-3}$ |
> | 3 | $6.6933 \times 10^{-5}$      | $3.1740 \times 10^{-3}$      | $< 1.0000 \times 10^{-7}$    | $5.5 \times 10^{-3}$ |
> | 5  | $5.8387 \times 10^{-5}$      | $3.1740 \times 10^{-3}$      | $< 1.0000 \times 10^{-7}$    | $7.3 \times 10^{-3}$ |
>
> We can observe that the average estimation error is approximately within $1 \times 10 ^{-4}$. Next, we evaluate the SVD performance based on $R_{\text{acc}}$ for different estimation accuracies at this error level. Here, the estimation error is artificially introduced by adding Gaussian noise to the estimated singular value. This is based on our observation that such noise has a uniform effect on singular values, regardless of their magnitude. Specifically, we define the singular value matrix with Gaussian noise as $S_{noise}(b) = S + bE$, where each component $E_{ij} \sim N(0,1)$ represents Gaussian noise. The results show that our method exhibits robust performance against estimation error.
>
> **Table 3: The Performance of Singular Value Estimation under Different Noise Levels**
>
> | Data    | m,n  | k   | b = 0  | b = $1 \times 10^{-5}$ | b = $1 \times 10^{-4}$ | b = $1 \times 10^{-3}$ |
> | ----------------------------------------- | --------- | --- | ------ | ---------------------- | ---------------------- | ---------------------- |
> | Baboon.png | 256,256   | 100 | 0.9072 | 0.9072 | 0.90710  | 0.9003   |
> | Hill.png  | 512,512   | 100 | 0.9612 | 0.9611  | 0.9578    | 0.9511 |
> | Matrix with power decay $\alpha=0.5$      | 100,100   | 100 | 0.9979 | 0.9978| 0.9978    | 0.9965    |
> | Matrix with exponential decay $\beta=0.5$ | 1000,1000 | 250 | 0.9999 | 0.9999   | 0.9999       | 0.9998          |
>
>
> ### 2. For the adaptive mechanism for Des-SVD
>
> Following your valuable suggestion, we incorporate a robustness mechanism to monitor the computation process. Specifically, we will check whether all variables become zero or if the objective function value turns negative. In such cases, we will terminate the calculation.
>
> Regarding the adaptive mechanism, our current experimental results indicate that if the initial singular value estimation contains significant errors, it is challenging to correct them effectively. In our future work, we will carefully consider your valuable suggestion and explore how to adaptively adjust the parameter $\gamma$ during the optimization iteration process.

---

> > ### Author Response · Authors · 2025-11-20
> > **Response to Reviewer bebZ (2/4)**
> >
> > **Choice and construction of D (Q2)**
> >
> > Thank you for your attention to the compatible matrix $D$. The construction of the compatible matrix $D$ is discussed in **Corollary 3** of [1]. Specifically, for SVD, the compatible matrix is the pseudo-inverse or the inverse of $A$. When randomization is applied, the target matrix is reduced to a $k \times k$ matrix, and $D$ can then be obtained by inversion, with a computational cost of $O(k^3)$. Overall, the time cost associated with this step is negligible in the entire process.
> >
> > An example is reported here. We test the time cost of each part of Des-SVD on two types of data, and the detailed results are shown in **Table 4**.
> >
> > **Table 4: Time Cost Breakdown of Des-SVD** (The slowest stage is **bolded**, and the second slowest stage is *italicized*.)
> >
> > | Time Stage  | Hill.png ($512 \times 512$), k = 150 | $100 \times 100$ matrix with power decay $\alpha = 0.5$, k=100 |
> > | ----------------------------------- | ------------------------------------ | -------------------------------------------------------------- |
> > | Randomized subspace iteration | 0.1370s | 0.0087s |
> > | Rayleigh quotient iterations  | 0.0096s  | 0.0029s   |
> > | Construction of compatible matrix   | 0.0069s | 0.0024s |
> > | Initialization of shared memory     | **0.3663s**   | *0.0651s* |
> > | Newton method     | 0.2207s   | 0.0516s  |
> > | Communication in parallel execution | *0.2384s* | **0.1234s**  |
> > | **Total** | 0.9789s    | 0.2541s    |
> >
> >  We can easily observe that the construction of the compatible matrix takes less than **1%** of the total execution time of Des-SVD, which aligns with our expectations.
> >
> > ----
> >
> > **Newton solver and ablations (Q3)**
> >
> > Thank you for your questions regarding the details of the Newton method. We would like to address them in the following two points respectively.
> >
> > 1. We adapt `torch.linalg.solve()` function to solve the linear fucntion and we have regularization on the KKT matrix to avoid singularity, and we will an adaptive mechanism following your sincere suggestion to handle ill-conditioned scenarios in the future.
> > 2. For the ablation study of the parameters in the Newton method, we conduct the following experiment. We select a $100 \times 100$ matrix with exponential decay $\beta = 0.5$. Several groups of common parameters are chosen, i.e. $n_{\text{max}} \in \{1, 3, 5, 10\}$ and $(\alpha, \beta) \in \{(0.1, 0.8), (0.01, 0.99), (0.2, 0.7), (0.5, 0.5)\}$. The ablation experiment results are shown in **Table 5**.
> >
> > **Table 5: Ablation Study of Newton Method Parameters**
> >
> > | **$n_{\text{max}}$** | **$\alpha$** | **$\beta$** | **Accuracy** | **Time (sec)** |
> > | -------------------- | ------------ | ----------- | ------------ | --------------- |
> > | 1                    | 0.1          | 0.8         | 0.9952       | 0.1137          |
> > | 3                    | 0.1          | 0.8         | 0.9952       | 0.1507          |
> > | 5                    | 0.1          | 0.8         | 0.9952       | 0.1782          |
> > | 10                   | 0.1          | 0.8         | 0.9952       | 0.2579          |
> > | 1                    | 0.01         | 0.99        | 0.9952       | 0.4233          |
> > | 3                    | 0.01         | 0.99        | 0.9952       | 1.0242          |
> > | 5                    | 0.01         | 0.99        | 0.9952       | 1.6732          |
> > | 10                   | 0.01         | 0.99        | 0.9952       | 3.2992          |
> > | 1                    | 0.2          | 0.7         | **0.9952**   | **0.0871**      |
> > | 3                    | 0.2          | 0.7         | 0.9952       | 0.1256          |
> > | 5                    | 0.2          | 0.7         | 0.9952       | 0.2340          |
> > | 10                   | 0.2          | 0.7         | 0.9952       | 0.2878          |
> > | 1                    | 0.5          | 0.5         | 0.9952       | 0.1182          |
> > | 3                    | 0.5          | 0.5         | 0.9952       | 0.1608          |
> > | 5                    | 0.5          | 0.5         | 0.9952       | 0.1769          |
> > | 10                   | 0.5          | 0.5         | 0.9952       | 0.2543          |
> >
> > **Hyper-parameters**: $n_{\text{max}} = 3$, $\alpha = 0.2$, $\beta = 0.7$.
> >
> > Overall, the performance is not highly sensitive to the backtracking parameters, although there are some small differences. We recommend using $\alpha = 0.2$ and $\beta = 0.7$, which are the values we use in all of our experiments. Additionally, we set $n_{\text{max}} = 3$. While we cannot theoretically claim that 3 iterations guarantee convergence, the accuracy achieved with this setting is sufficient to provide an accurate SVD.

---

> > > ### Author Response · Authors · 2025-11-20
> > > **Response to Reviewer bebZ (3/4)**
> > >
> > > **Extensions and limitations (Q4)**
> > >
> > > We sincerely thank you for your constructive suggestions. We would like to address them respectively as follows.
> > >
> > > **For extensions**
> > >
> > > In fact, our parallelization method can easily be adapted to block methods, which calculate multiple blocks simultaneously. However, when singular values are clustered and too similar, our method may encounter issues, as described below. In such cases, we often treat them as a single singular value to mitigate these problems.
> > >
> > > **For limitations**
> > >
> > > We have evaluated the failure modes by examining scenarios that have exhibited notable singular-value estimation errors (see **Singular Value Estimation and Accuracy (Q1)**). Our analysis has further indicated that the method may fail when singular values have lain extremely close to one another. To handle these special cases, we adopted the strategy described in Appendix Section E. Specifically, when the gap satisfies $ε < 1 \times 10^{-3}$, we regard the singular values as effectively identical because their differences become numerically insignificant. Accordingly, for such clustered singular values, we treat them as identical and compute only a single representative value.
> > >
> > > ----
> > >
> > > **Theoretical clarifications (Q5)**
> > >
> > > Thank you for your questions regarding the theoretical aspects. We will address each point separately.
> > >
> > > 1. We appreciate your thoughtful suggestion. As described in **Singular Value Estimation and Accuracy (Q1)**, we will check whether all variables become zero or if the objective function value turns negative. In such cases, we will terminate the calculation and restart the singular-value estimation to prevent failure and ensure stability.
> > >
> > >
> > > 2. Thank you for your insightful recommendation. We will consider this approach in future work. Incorporating data-dependent methods would provide a better understanding of the boundaries for this estimation, which we believe could enhance our approach.
> > >
> > >
> > > ---
> > >
> > > **Reproducibility and code (Q6).**
> > >
> > > Thank you for your suggestion. We report the parameter settings we used one by one and will include them in the revised version of our manuscript.
> > >
> > > - For the randomized subspace iteration, we follow Algorithm 4.4 in [2] to compute the orthonormal matrix $Q$, and we set the number of iterations to $q = 5$.
> > >
> > > - For the Rayleigh quotient iteration, as mentioned in **Singular Value Estimation and Accuracy(Q1)** , we set $n_{iter} = 3$ and the tolerance $\epsilon = 1e-6$.
> > >
> > > - Regarding the line search strategy, as discussed in **Newton solver and ablations(Q3)**, we use $n_{\text{max}} = 3$, $\alpha = 0.7$, and $\beta = 0.2$ in practice.
> > >
> > > ----
> > >
> > > **Complexity and Scaling Details (W1)**
> > >
> > > We appreciate your interest in the time complexity of our method. For the time cost of each stage, please refer to **Table 4** as mentioned in **Choice and construction of D(Q2)**. For a more detailed breakdown of the time complexity for each stage, please see **Table 6**.
> > >
> > > **Table 6: Time Complexity of Each Stage**
> > >
> > > | Stage                               | Time Complexity |
> > > | ----------------------------------- | --------------- |
> > > | Randomized subspace iteration       | $O(mnk)$        |
> > > | Rayleigh quotient iterations        | $O(k^3)$        |
> > > | Construction of compatible matrix   | $O(k^3)$        |
> > > | Newton method                       | $O((6k)^3)$     |
> > >
> > > As observed, after dimensionality reduction via the Randomized Subspace Iteration, the complexity of our algorithm is reduced to $O((6k)^3)$.
> > >
> > > ----
> > >
> > > **About Normalization (W2)**
> > >
> > > Thank you for your thoughtful suggestion. Since normalization does not affect the satisfaction of the KKT conditions, we have chosen to apply it only after all iterations. This approach ensures efficiency without negatively impacting the final results or the overall convergence of the algorithm. We will include this clarification in the revised manuscript.

---

> > > > ### Author Response · Authors · 2025-11-20
> > > > **Response to Reviewer bebZ (4/4)**
> > > >
> > > > **Comparisons and Breadth (W3)**
> > > >
> > > > Thank you for your suggestion regarding other dense SVD methods and parallelization on GPU. We will address these points accordingly.
> > > >
> > > > 1. Thank you for your reminder. The state-of-the-art method we compare against is Lan-SVD, which is a more efficient and commonly used algorithm compared to Jacobi. Moreover, the Lanczos method is essentially an accelerated form of subspace iteration and is often faster than power iteration. Here, we include a comparison with power iteration to demonstrate the high efficiency of Lan-SVD. We select small matrices to complete the full SVD as a simulation of the matrix after randomization. The results in **Table 7** support our conclusion.
> > > >
> > > > **Table 7: Time Consumption of Power Iteration and Lan-SVD**
> > > >
> > > > | m, n     | Power Iteration | Lan-SVD |
> > > > | -------- | ---------------- | ------- |
> > > > | 50, 50   | 0.2307s          | 0.0388s |
> > > > | 100, 100 | 0.5522s          | 0.0996s |
> > > > | 200, 200 | 0.8024s          | 0.2850s |
> > > >
> > > > 2. Thank you for your question. We would like to emphasize that the key advantage of our algorithm lies in its ability to be parallelized, enabling acceleration independent of the device used. We have already compared the impact of different cores/nodes on speedup, and the details are provided in **Table 8**. We appreciate your suggestion, and integrating our method with lower-level GPU parallel computation is indeed an important direction for future development, which would involve engineering customizations of low-level kernel operators.
> > > >
> > > > **Table 8: Parallel Performance on Baboon.png (256 x 256)**
> > > >
> > > > | k   | Sequential | Parallel | Speedup |
> > > > | --- | ---------- | -------- | ------- |
> > > > | 10  | 0.2207s    | 0.141s   | 1.57    |
> > > > | 50  | 21.247s    | 0.505s   | 42.07   |
> > > > | 100 | 135.072s   | 1.209s   | 111.66  |
> > > >
> > > > ----
> > > >
> > > > **Theoretical scope (W4)**
> > > >
> > > > The raised problems are indeed very important, and some of them are fundamentally challenging due to the non-convexity of the underlying formulation. Issues such as the basin of attraction and global convergence lie at the core of non-convex optimization, and obtaining a general characterization is inherently difficult.
> > > >
> > > > For Des-SVD in particular, we have shown that when the singular value estimation error falls below the threshold $T_{err}$, we can guarantee recovery of the correct decomposition. Under this condition, the convergence behavior follows the standard Newton-type analysis. However, once the error exceeds $T_{err}$, the analysis becomes significantly more challenging (as discussed in **Singular Value Estimation and Accuracy (Q1)**). Your suggestion is especially insightful: by adapting the step size parameter $\gamma$ to ensure that the objective value does not become negative, the method may become substantially more stable and better able to tolerate larger estimation errors.
> > > >
> > > > We will add the following discussion at the end of the manuscript:
> > > >
> > > > > As stated in Theorem 3.2, a key condition for Des-SVD to obtain the true singular vectors is that the singular value estimation satisfies the threshold $T_{err}$. When this condition is met, the convergence follows the standard behavior of the Newton method. Otherwise, the objective value may become negative, indicating a failure of the decomposition. An interesting direction for future research is to investigate how to adapt or modify the singular value estimation—potentially improving the robustness of Des-SVD and enabling stability even when the estimation error exceeds the current threshold.
> > > >
> > > > ----
> > > >
> > > > **Overall Reply**
> > > >
> > > > We sincerely appreciate your thoughtful and constructive suggestions on our work. In response, we will revise and refine our manuscript in the following three major aspects:
> > > >
> > > > - Theoretical Scope: We will further clarify the robustness (**Q1**) and convergence properties (**W4**) of our method.
> > > >
> > > > - Algorithmic Details: We will provide additional explanations regarding normalization (**W2**), the robustness mechanism (**Q1**), RSVD parameter choices (**Q6**), and other related components.
> > > >
> > > > - Supplementary Experiments: We will include ablation studies on Newton iterations (**Q3**), scaling experiments demonstrating parallel efficiency (**W3**), and a detailed breakdown of the time cost of Des-SVD (**Q2**), among others.
> > > >
> > > > We hope that these improvements will make our work more complete, clearer, and more practically useful. We sincerely thank you again for your valuable and insightful feedback.
> > > >
> > > > ---
> > > > **References**:
> > > >
> > > > [1] Suykens, Johan AK. "SVD revisited: A new variational principle, compatible feature maps and nonlinear extensions." Applied and Computational Harmonic Analysis 40.3 (2016): 600-609.
> > > >
> > > > [2] Halko, N., Martinsson, P.G. and Tropp, J.A., 2011. Finding structure with randomness: Probabilistic algorithms for constructing approximate matrix decompositions. SIAM review, 53(2), pp.217-288.

---

### Official Review · Reviewer_6HC2 · 2025-11-02

**Soundness:** 2
**Presentation:** 3
**Contribution:** 2
**Rating:** 2
**Confidence:** 3

**Summary:**

### Summary

This paper introduces "Des-SVD," a new algorithm for Singular Value Decomposition (SVD), motivated by the need for methods that are more amenable to parallelization than traditional matrix-based algorithms and more efficient than existing descent-based approaches (like Riemannian gradient descent).

The proposed solution is based on a primal-dual reformulation (first proposed by Suykens, 2016) that connects SVD to the KKT conditions of a non-convex least-squares problem.

The paper's core contributions are:
1.  **Theoretical Analysis (Section 3):** A key insight (Theorem 3.1) that for this non-convex problem, non-zero KKT solutions—which can be normalized to yield singular vectors—*only* exist when the correct singular value $s$ is used as a parameter. An incorrect $s$ leads to a trivial zero solution.
2.  **A New Algorithm (Des-SVD):** A practical, multi-stage algorithm that first estimates singular values (e.g., via Rayleigh-Quotient Iteration) and then uses this theoretical insight to solve for the singular vectors using an iterative method (Algorithm 1), which is shown to be Newton's method on the KKT system.
3.  **Empirical Results:** The paper demonstrates that Des-SVD is "significantly higher" in efficiency than Riemannian SVD and "competitive with" (or even faster than) standard matrix-based methods like Jacobi-SVD and Lanczos-SVD.

**Strengths:**

1.  **Novel Theoretical Insight:** The paper's core theoretical contribution (Theorem 3.1), which analyzes the KKT conditions of the (Suykens, 2016) formulation, is a valuable and novel insight. It provides a clear theoretical justification for *why* a non-convex least-squares problem can be used to find the SVD, showing that non-trivial solutions are only possible when the correct singular value is used as a parameter.
2.  **Strong Empirical Performance:** The proposed Des-SVD algorithm demonstrates impressive empirical results. It is shown to be significantly more efficient than the primary descent-based alternative (Rie-SVD) and performs competitively against, or even faster than, standard matrix-based methods like Lanczos-SVD on several benchmarks.
3.  **Valid Problem Motivation:** The paper correctly identifies a valid gap in the literature: the desire for SVD algorithms that are better suited for modern, large-scale, and potentially distributed/parallel computing paradigms, where traditional centralized matrix methods can be bottlenecks.

**Weaknesses:**

### 1. Misleading Theoretical Framing: "Descent Method" vs. "Matrix-Based Method"

The paper's entire narrative is built on a false dichotomy: "matrix-based methods" vs. "descent methods." It then frames Des-SVD as a "descent method." This is fundamentally incorrect. The core of the "descent method" (Algorithm 1) is **Newton's method**, which requires solving a large, dense KKT linear system in each iteration. This is a "matrix-based operation" by any definition. The paper's own complexity analysis (Section 4.2) states this KKT system is $(6k \times 6k)$ and solving it is an $O((6k)^3)$ operation. The paper simply **replaces one set of matrix operations** (e.g., Lanczos) with **another** (Newton on a KKT system).

### 2. The Core SVD Problem is Assumed Solved

The paper's "novel" part, Algorithm 1, is titled "Descent method for calculating the singular vectors **from a given singular value**." This framing reveals the core weakness: the method *presupposes* that the singular values ($s$) are already known. Algorithm 2 (the "refined" version) is a **multi-stage, hybrid, matrix-based algorithm** that uses classical matrix methods (like Randomized-Subspace-Iteration and Rayleigh-Quotient Iteration) to find $s$ *before* the novel part of the algorithm ever runs.

### 3. Lack of Robustness Analysis and Convergence Guarantees

The algorithm's structure introduces new questions that the paper leaves unanswered.
* **Missing Robustness Analysis:** The algorithm's success hinges on an accurate estimate of $s$. While a theoretical bound ($T_{err}$) is provided (Theorem 3.2), the **empirical impact of a misspecified $s$ is never shown**.
* **Missing Convergence Guarantees:** For a new numerical algorithm, the paper provides **no explicit convergence guarantees** for the Newton solve in Algorithm 1.

### 4. Unsubstantiated Claims of Parallelization

The paper's primary motivation is parallelization, but its claims are weak and unproven.
* The claim that matrix-based methods are "unfriendly to parallelization" is a strong and largely inaccurate generalization.
* The paper's "parallelization" (Algorithm 2, Step 11) is an "embarrassingly parallel" loop over $k$ values, which is only possible *after* a **serial, matrix-based pre-computation** (Step 8) has found all $k$ singular values.
* The paper provides **no empirical evidence** (e.g., a scaling plot) to quantify its parallel performance.

### 5. Incremental Novelty

The core primal-dual formulation (Equation 4) is explicitly attributed to **Suykens (2016)**. The paper's *true* theoretical contribution is the *analysis* of this existing formulation, making the novelty more incremental than presented.

**Questions:**

I may have missed this but what is p in table 2?

---

> ### Author Response · Authors · 2025-11-20
> **Response to Reviewer 6HC2 (1/3)**
>
> We sincerely thank you for your constructive and valuable comments. Accordingly, we address each of your concerns in detail as follows.
>
> ----
>
> **"Descent Method" vs "Matrix-based Method"(W1)**
>
> We understand your concern, but we still think "the Newton method is a kind of descent method". This statement is not specific to our problem but rather a well-accepted concept. It is true that the Newton method involves matrix operations, but the matrix operations are used to calulate the descent direction and to update the solution.
>
> In our view, the key distinction between "descent methods" and "matrix-based methods" lies in the presence of an objective function to minimize. When such an objective exists, the goal of a descent method is to identify a feasible descent direction (which, in most cases, involves matrix operations) and iteratively update the solution to reach the minimum.
>
> Thanks for your suggestion and we will further clarify this key point in the revised manuscript.
>
> ----
>
> **Core SVD Problem is Assumed Solved (W2)**
>
> Thanks for your question regarding the singular value estimation. First of all, estimating the singular value is much easier than finding the singular vectors. In high-level terms, estimating a singular value is a univariate problem, while finding its singular vectors deals with an $n$-dimensional vector. In practice, we apply Rayleigh-Quotient Iteration, which is fast and accurate enough for Des-SVD. In the following table, we detail the computational time of each step in Des-SVD.
>
> **Table 1: Time Cost Breakdown of Des-SVD** (the slowest stage is **bolded**, and the second slowest stage is *italicized*)
>
> | Time Stage                          | Hill.png ($512 \times 512$), k = 150 | $100 \times 100$ matrix with power decay $\alpha = 0.5$, k = 100 |
> | ----------------------------------- | ----------------------------------- | -------------------------------------------------------------- |
> | Randomized subspace iteration    | 0.1370s  | 0.0087s  |
> | Rayleigh quotient iterations        | 0.0096s| 0.0029s  |
> | Construction of compatible matrix   | 0.0069s  | 0.0024s |
> | Initialization of shared memory     | **0.3663s**  | *0.0651s*    |
> | Newton method                       | 0.2207s | 0.0516s |
> | Communication in parallel execution | *0.2384s*   | **0.1234s**  |
> | **Total**                           | 0.9789s | 0.2541s  |
>
> One can observe that the Rayleigh quotient iteration accounts for only about $1\%$ of the total execution time of Des-SVD. Thus, estimating the singular values does not indicate that the full SVD has already been solved; rather, it requires only a negligible amount of additional computation.

---

> > ### Author Response · Authors · 2025-11-20
> > **Response to Reviewer 6HC2 (2/3)**
> >
> > **Robustness analysis and convergence guarantees (W3)**
> >
> > Thanks a lot for the questions about robustness and convergence.
> >
> > Regarding convergence, when Des-SVD correctly solves the SVD, the convergence follows the standard Newton method. If it does not converge, the objective value rapidly becomes negative, which provides a clear signal to terminate the algorithm.
> >
> > As for robustness, several factors may be considered. We have evaluated the behavior of the method under different singular value decay rates and varying ranks, and we have also examined the case where two singular values are close to each other (as detailed in **Appendix E**). Overall, the robustness of Des-SVD is comparable to that of the Lanczos method.
> >
> > A specific robustness issue in Des-SVD is the singular value estimation. We first evaluate the performance of the Rayleigh method using different numbers of iterations.
> >
> > In the following, we use $Err_{avg}(S_{es})$ to describe the average estimation error.  Let $S$ denote the true singular value matrix and $S_{es}$ the estimated one. We define the average estimation error as:
> >
> > $$
> > Err_{avg}(S_{es}) = \frac{1}{k} \| S_{es} - S \|_F,
> > $$
> >
> > where $k$ is the number of singular values. Experiments in **Table 2** and **Table 3** show that the Rayleigh iteration method converges effectively, and we choose $n_{iter} = 3$ for all reported experiments. We also report the maximum and minimum values of estimation error across all singular values to demonstrate that the estimation error is well-balanced and has minimal impact on different singular values.
> >
> > **Table 2: Rayleigh Iteration Performance on Hill.png**
> >
> > **Data**: Hill.png, $k = 100$.
> >
> > | $n_{iter}$ | $Err_{avg}(S_{rayleigh})$ | $Err_{max}(S_{rayleigh})$ | $Err_{min}(S_{rayleigh})$ | Time (sec)           |
> > | ---------- | ------------------------- | ------------------------- | ------------------------- | -------------------- |
> > | 1          | $3.5864 \times 10^{-7}$   | $1.6000 \times 10^{-5}$   | $< 1.0000 \times 10^{-7}$ | $3.7 \times 10^{-3}$ |
> > | 3          | $2.1186 \times 10^{-7}$   | $1.1000 \times 10^{-5}$   | $< 1.0000 \times 10^{-7}$ | $5.6 \times 10^{-3}$ |
> > | 10         | $1.8267 \times 10^{-7}$   | $1.0000 \times 10^{-5}$   | $< 1.0000 \times 10^{-7}$ | $1.2 \times 10^{-2}$ |
> >
> > **Table 3: Rayleigh Iteration Performance on Matrix with Exponential Decay**
> >
> > **Data**: Matrix(100 $\times$ 100) with power decay $\alpha = 0.5$, $k = 100$.
> >
> > | $n_{iter}$ | $Err_{avg}(S_{rayleigh})$ | $Err_{max}(S_{rayleigh})$ | $Err_{min}(S_{rayleigh})$ | Time (sec)           |
> > | ---------- | ---------------------------- | ---------------------------- | ---------------------------- | -------------------- |
> > | 1          | $6.8593 \times 10^{-5}$      | $3.6620 \times 10^{-3}$      | $< 1.0000 \times 10^{-7}$    | $4.2 \times 10^{-3}$ |
> > | 3          | $6.6933 \times 10^{-5}$      | $3.1740 \times 10^{-3}$      | $< 1.0000 \times 10^{-7}$    | $5.5 \times 10^{-3}$ |
> > | 5          | $5.8387 \times 10^{-5}$      | $3.1740 \times 10^{-3}$      | $< 1.0000 \times 10^{-7}$    | $7.3 \times 10^{-3}$ |
> >
> > We can observe that the average estimation error is approximately within $1 \times 10 ^{-4}$. Next, we evaluate the SVD performance based on $R_{\text{acc}}$ for different estimation accuracies at this error level. Here, the estimation error is artificially introduced by adding Gaussian noise to the estimated singular value. This is based on our observation that such noise has a uniform effect on singular values, regardless of their magnitude. Specifically, we define the singular value matrix with Gaussian noise as $S_{noise}(b) = S + bE$, where each component $E_{ij} \sim N(0,1)$ represents Gaussian noise. The results show that our method exhibits robust performance against estimation error.
> >
> >
> > **Table 4: The Performance of Singular Value Estimation under Different Noise Levels**
> >
> > | Data    | m,n   | k   | b = 0  | b = $1 \times 10^{-5}$ | b = $1 \times 10^{-4}$ | b = $1 \times 10^{-3}$ |
> > | ----------------------------------------- | --------- | --- | ------ | ---------------------- | ---------------------- | ---------------------- |
> > | Baboon.png  | 256,256   | 100 | 0.9072 | 0.9072 | 0.90710| 0.9003  |
> > | Hill.png  | 512,512   | 100 | 0.9612 | 0.9611 | 0.9578 | 0.9511 |
> > | Matrix with power decay $\alpha=0.5$      | 100,100   | 100 | 0.9979 | 0.9978| 0.9978  | 0.9965   |
> > | Matrix with exponential decay $\beta=0.5$ | 1000,1000 | 250 | 0.9999 | 0.9999 | 0.9999  | 0.9998 |

---

> > > ### Author Response · Authors · 2025-11-20
> > > **Response to Reviewer 6HC2 (3/3)**
> > >
> > > **Claims of Parallelization (W4)**
> > >
> > >
> > > Thanks for your questions regarding parallelization. We will respond to each of them below.
> > >
> > > - Regarding the claim that matrix-based methods are "unfriendly to parallelization", we have reviewed several parallelized SVD methods , many of which face communication or data privacy issues due to the inability to solve subproblems independently. Upon reflection, we now realize that we should not arbitrarily claim that "matrix-based methods are unfriendly to parallelization." Instead, we will restrict the discussion to SVD and rephrase the claim as: **"Matrix-based methods, compared to descent methods, are relatively less friendly to parallelization."**
> > >
> > >
> > > - Regarding the singular value estimation step, we treat it as a "preconditioning" step rather than part of the core method that involves parallelization (see more in **Core SVD Problem is Assumed Solved (W2)**). We sincerely appreciate your thoughtful suggestions and will incorporate this clarification in the revised version of our paper.
> > >
> > > - Regarding the parallel performance details, we conducted a supplementary experiment on the common image Baboon.png ($256 \times 256$) to demonstrate the speedup achieved by our parallelization approach. As shown in Table 3, when $k$ is small, the speedup is not significant. However, as $k$ increases, a considerable speedup is observed. This can be attributed to the fixed overhead of operations such as shared memory preparation, which is necessary for parallel execution. Therefore, the benefits of parallelization become more pronounced with larger-scale computations.
> > >
> > >     **Table 5: Parallel Performance on Baboon.png**
> > >
> > >     | k   | Sequential | Parallel | Speedup |
> > >     | --- | ---------- | -------- | ------- |
> > >     | 10  | 0.221s     | 0.141s   | 1.6     |
> > >     | 50  | 21.247s    | 0.505s   | 42.1    |
> > >     | 100 | 135.072s   | 1.209s   | 111.7   |
> > >
> > > ----
> > >
> > > **Contribution from Suykens (2016) (W5)**
> > >
> > >
> > > Thank you for your question regarding the novelty of our work.
> > >
> > > The primal–dual relationship introduced by Suykens (2016) is indeed a fundamental work that offers a completely different perspective—an optimization view—on investigating SVD. We sincerely appreciate you for highlighting this important connection. At first glance, such a primal–dual relationship may appear to naturally suggest a descent method, but in practice it is far more challenging than it seems, which explains why little progress has been made in this direction over the past decade.
> > >
> > > The core difficulty lies in the non-convexity of the primal problem, which leads to many minima that do not correspond to the true SVD, thus hindering the development of effective algorithms. Our main contribution directly addresses this challenge: we have theoretically proven and empirically validated that our approach guarantees recovery of the correct SVD. While the underlying technique may not appear overly sophisticated, its contribution is both crucial and practical—it enables a descent method that is substantially more efficient than all previous descent-based SVD approaches.
> > >
> > > ----
> > >
> > > **A Typo in Table 2 (Q1)**
> > >
> > > Thanks a lot for pointing out our typo. It should be $k$, the number of singular values we need. We will certainly update this in the revised version.

---

### Official Review · Reviewer_nBDS · 2025-11-03

**Soundness:** 2
**Presentation:** 3
**Contribution:** 2
**Rating:** 6
**Confidence:** 3

**Summary:**

A descent method is developed based on the LS formulation and its dual. Basic properties of the formulation are briefly discussed.

**Strengths:**

A descent method based on the LS formulation, and the discussion of the basic properties. The idea is simple and interesting.

**Weaknesses:**

I am not sure whether the method is really practical or not. Indeed, it is not compared with the true SOTA solvers for SVD. For dense matrices, the QR algorithm and the divide and conquer algorithm are more efficient. For sparse matrices, there also exists iterative optimization solver such as LOBPCG which is well developed. In addition, the sizes of some test problems are relatively small.

**Questions:**

See weakness.

---

> ### Author Response · Authors · 2025-11-20
> **Response to Reviewer nBDS**
>
> We sincerely thank you for your constructive and valuable comments. Accordingly, we address each of your concerns in detail as follows.
>
> ---
>
> **QR Solver (W1)**
>
> Thank you for bringing up other SVD methods. For dense matrices, Lan-SVD and Jac-SVD are the most popular ones. For example, the `torch.linalg.svd` function uses Jacobi SVD for implementation.
>
> For the QR method, its theoretical complexity is $O(n^3)$, which is the same as Lanczos and Jacobian methods. Similarly, the Divide and Conquer method also has a complexity of $O(n^3)$. However, Lan-SVD is more versatile and scalable, which is why it is more commonly used in modern applications.
>
> ---
>
> **LOBPCG (W2)**
>
> Thanks for mentioning LOBPCG.
>
> LOBPCG is designed to efficiently compute the top few eigenvalues and eigenvectors of sparse and symmetric matrices. Its potential limitations in the context of SVD are summarized below:
>
> 1. **Pre-calculation Requirement**: Applying LOBPCG to the SVD problem requires precomputing $A A^T$ in order to perform eigen decomposition.
>
> 2. **Top-k Limitation**: LOBPCG is tailored for computing only the top-$k$ eigenvalues and is not suitable when a larger portion of the spectrum is required.
>
> 3. **Sparsity Considerations**: While LOBPCG performs well on sparse matrices, its accuracy and efficiency degrade as the matrix becomes denser.
>
> 4. **Difficulty with Random Sampling**: After random sampling, the resulting blocks remain nearly as large as the reduced matrix, making sparsity-based methods ineffective.
>
> To validate these observations, we conduct a simple experiment comparing the performance of LOBPCG and Lan-SVD under varying matrix sparsity. The results are presented in **Table 1**.
>
> **Table 1: Comparison of Lan-SVD and LOBPCG**
>
> | m,n,k           | Density | Time in sec (Lan-SVD) | Time in sec (LOBPCG) |
> | --------------- | ------- | --------------------- | -------------------- |
> | 1000,1000,100   | 0.01    | 0.0638                | 0.1475               |
> | 1000,1000,100   | 0.5     | 0.1267                | 0.3366               |
> | 20000,20000,100 | 0.01    | 2.0489                | 3.6783               |
> | 20000,20000,100 | 0.5     | 2.1726                | 8.1759               |
>
> From **Table 1**, we observe that Lan-SVD outperforms LOBPCG in terms of efficiency. In contrast, LOBPCG exhibits significantly lower efficiency as matrix density increases. This experiment provides insight into why Lan-SVD is a more commonly used SVD solver. Therefore, we have chosen Lan-SVD as our comparison method due to its efficiency across a wide range of scenarios. However, the performance advantage of LOBPCG with sparse matrices suggests that there is potential to optimize Des-SVD for sparse matrix scenarios, and we will incorporate this in future work.
>
> ---
>
> **Scale of the Experiments (W3)**
>
> Thank you for your insight regarding the scale of our experiments. In the current experiments, we have tested different problem scales, with the largest size being $4000 \times 6000$, which covers most real-world images.
>
> Regarding scalability, you may be thinking of Large Language Models (LLMs), where SVD is also widely used. Typical scenarios include LoRA initialization and low-rank approximation of the attention mechanism. For such problems, matrix sizes typically range from tens of thousands (since common QKV matrix/FFN weight matrix sizes are usually in the tens of thousands).
>
> To further address your concerns, we have conducted additional validation in a 10,000 × 10,000 scenario. We simulated a matrix with size 10,000 × 10,000 using exponential decay ($\beta=0.1$). The detailed results are shown in **Table 2**, which illustrates that our method still performs well on large-scale matrices.
>
> **Table 2: Performance of Des-SVD and Lan-SVD on Large Matrices (10,000 × 10,000)**
> | $k$ | $R_\text{acc}$ (Des-SVD) | $R_\text{acc}$ (Lan-SVD) | Time in sec(Des-SVD) | Time in sec(Lan-SVD) |
> | --- | ------------------------ | ------------------------ | -------------------- | -------------------- |
> | 50  | 0.9932                   |    0.9932                      | 1.9705              |   1.8121                   |
> | 100 | 0.9993                   |     0.9993                     |  3.1097             |     2.8416                 |

---

### Official Review · Reviewer_ZVWz · 2025-11-04

**Soundness:** 3
**Presentation:** 3
**Contribution:** 3
**Rating:** 6
**Confidence:** 3

**Summary:**

The work revisits the classical singular value decomposition problem and aims to propose a computationally efficient algorithm based on gradient descent. The authors build their work on the result that nonzero KKT solutions of a least squares primal problem give the singular vectors under certain conditions. The method can support parallelization unlike the classical approaches and hence can reduce the computational time compared to projection-based approaches like Riemann SVD. The experiments are presented to showcase the effectiveness of the approach.

**Strengths:**

Strengths:

1.	The paper is written in an organized, well-articulated manner. It is easy to follow (in most parts) and the theoretical results are discussed in sufficient detail.

2.	The problem statement is also relevant and timely as decomposition methods are recently getting a lot of attention in AI-based efficient/interpretable models including in the domain of LLMs

**Weaknesses:**

Weaknesses/Questions:

1.	Problem (4) seems like the primal problem. It is mentioned before 4 that it is the dual formulation.

2.	The dual solutions $\alpha$ and $\beta$ are introduced and discussed before stating the dual problem which brings some confusion and less clarity even in the notations itself.

3.	In Section 3.2, it is intended to show that when the singular values are not known to set the regularization coefficient, we get zero solutions. It is not clear how does the KKT matrix is shown to be full rank. It would be better to give some insights of this proof step since it is very important in the message conveyed in Section 3.2.

4.	In Section 3.2 itself, $\bm v$ is mentioned as the Lagrange operator initialized as 0 and also the primal variables. So it is unclear how you obtained Eq. 17 from Eq. 16. There is some lack of clarity in these steps.

5.	I think, the proof does not show there exists no other nonzero KKT solutions that are not singular vectors.

6.	In the practical implementation of the algorithm, the inputs require C matrix which needs the mapping functions to be computed that involves an estimation of D and also the prior estimation of singular values. Do you also consider these computational costs when comparing them with other algorithms? If not, I highly suggest the comparison has to be with overall cost of obtained SVD starting from the input matrix A without assuming any other prior info.

7.	It is also mentioned that the normalization of the dual variables are done in each iteration. But in pseudo code, it is performed only once after all the iterations. Why is such a design choice?

8. Also, in the pseudo code, a check to see if the solutions are zero is missing. If the singular values are not specified exactly, how do we inspect and detect the failure scenarios?

9.	In experiments, there should also be some experiments to validate Theorem 3.2 to see the effect of misspecification of singular values.

**Questions:**

See the weaknesses section.

---

> ### Author Response · Authors · 2025-11-20
> **Response to Reviewer ZVWz (1/2)**
>
> We sincerely thank you for your constructive and valuable comments. Accordingly, we address each of your concerns in detail as follows.
>
> ---
> **Expressions Details (Q1)**
>
> Thank you for pointing out this typo. Problem (4) is indeed the primal problem, and we clarify the role of Problem (4) and the associated notation as follows.
>
> - Problem (4) represents the *primal least-squares formulation*, and its *dual* is exactly the SVD problem stated in (5).
>
> - The dual variables $\boldsymbol{\alpha}$ and $\boldsymbol{\beta}$ are introduced earlier than the dual problem because they naturally arise in the KKT conditions used to derive the dual SVD formulation. In terms of notation, since the dual problem corresponds to the SVD, $\boldsymbol{\alpha_i}$ denotes the $i^{th}$ left singular vector, with the left singular matrix $[\boldsymbol{\alpha}] = [\boldsymbol{\alpha}_1, \dots, \boldsymbol{\alpha}_n]^{\top}$, and $\boldsymbol{\beta}_r$ denotes the $r^{\text{th}}$ right singular vector, with the right singular matrix $[\boldsymbol{\beta}] = [\boldsymbol{\beta}_1, \dots, \boldsymbol{\beta}_m]^{\top}$.
>
> We sincerely appreciate your valuable suggestion, and we will incorporate the necessary revisions into our manuscript, highlighting this clarification.
>
> ----
> **Section 3.2 Details (Q2)**
>
>
> Indeed, the rank property of the KKT matrix is crucial. A complete proof can be found in **Appendix B.1**, where we utilize linear transformations and the connection to the KKT conditions.
>
> Regarding your question about "Eq. 16 $\rightarrow$ Eq. 17": since $\Delta t$ is decoupled from $x$, we can simply set $\Delta t = 0$, which leads to a feasible solution as given by Eq. 17. We will provide further clarification of these details in our manuscript.
>
> For your question about "nonzero KKT solutions that are not singular vectors": in Section 3.1, we show that after normalization, all KKT solutions satisfy orthogonality. Therefore, any nonzero KKT solution, when normalized, is guaranteed to be a singular vector.
>
> ----
> **Algorithm details (Q3)**
>
> Thank you for your careful reading and helpful suggestions.
>
> In fact, normalization ensures orthogonality and does not affect the satisfaction of the KKT conditions. Therefore, in the algorithm design, we only need to satisfy orthogonality after all other conditions have been met, i.e., after all iterations are completed. We will include this clarification in the revised manuscript.
>
> Additionally, your suggestion on detecting failure scenarios is excellent. As long as the variables become all zero or the value of the objective function becomes negative, we will terminate the calculation. We will also update our manuscript to incorporate this improvement.
>
> ---
>
> **Experiments Details (Q4)**
>
> Thank you for your feedback regarding the experimental details. We have designed the corresponding experiments as follows.
>
> ### 1. Computational cost breakdown
>
> We would like to clarify that the reported times include the overall computation, such as the estimation of the singular values and the compatible matrix $D$. To emphasize that these pre-processing steps take a relatively short amount of time, we present the time cost breakdown for each stage in the tables below.
>
> **Table 1: Time Cost Breakdown of Des-SVD**
> (The slowest stage is **bolded**, and the second slowest stage is *italicized*.)
>
> | Time Stage | Hill.png ($512 \times 512$), k = 150 | $100 \times 100$ matrix with power decay $\alpha = 0.5$, k=100 |
> | ----------------------------------- | ----------------------------------- | -------------------------------------------------------------- |
> | Randomized subspace iteration  | 0.1370s  | 0.0087s   |
> | Rayleigh quotient iterations  | 0.0096s  | 0.0029s|
> | Construction of compatible matrix| 0.0069s| 0.0024s |
> | Initialization of shared memory | **0.3663s**| *0.0651s* |
> | Newton method| 0.2207s | 0.0516s |
> | Communication in parallel execution | *0.2384s* | **0.1234s**   |
> | **Total** | 0.9789s| 0.2541s |

---

> > ### Author Response · Authors · 2025-11-20
> > **Response to Reviewer ZVWz (2/2)**
> >
> > ### 2. The effect of misspecification of singular values
> >
> > Thanks for your considerate suggestion. We have conducted supplementary experiments to evaluate the performance under misspecified singular values. Since our method relies on Rayleigh iteration for singular value estimation, we report the performance of Rayleigh iteration with different numbers of iterations to illustrate the general error bound we oberved in practice to test the misspecified singular values. In addition, we further demonstrate the robustness of our method by injecting Gaussian noise into the singular value estimates within the prescribed error bound, in order to validate Theorem 3.2.
> >
> > In the following, we use $Err_{avg}(S_{es})$ to describe the average estimation error. Let $S$ denote the true singular value matrix and $S_{es}$ the estimated one. We define the average estimation error as:
> >
> > $$
> > Err_{avg}(S_{es}) = \frac{1}{k} \| S_{es} - S \|_F,
> > $$
> >
> > where $k$ is the number of singular values. Experiments in **Table 2** and **Table 3** show that the Rayleigh iteration method converges effectively, and we choose $n_{iter} = 3$ for all reported experiments. We also report the maximum and minimum values of estimation error across all singular values to demonstrate that the estimation error is well-balanced and has minimal impact on different singular values.
> >
> > **Table 2: Rayleigh Iteration Performance on Hill.png**
> >
> > **Data**: Hill.png, $k = 100$.
> >
> > | $n_{iter}$ | $Err_{avg}(S_{rayleigh})$ | $Err_{max}(S_{rayleigh})$ | $Err_{min}(S_{rayleigh})$ | Time (sec)           |
> > | ---------- | ------------------------- | ------------------------- | ------------------------- | -------------------- |
> > | 1  | $3.5864 \times 10^{-7}$   | $1.6000 \times 10^{-5}$   | $< 1.0000 \times 10^{-7}$ | $3.7 \times 10^{-3}$ |
> > | 3   | $2.1186 \times 10^{-7}$   | $1.1000 \times 10^{-5}$   | $< 1.0000 \times 10^{-7}$ | $5.6 \times 10^{-3}$ |
> > | 10| $1.8267 \times 10^{-7}$   | $1.0000 \times 10^{-5}$   | $< 1.0000 \times 10^{-7}$ | $1.2 \times 10^{-2}$ |
> >
> > **Table 3: Rayleigh Iteration Performance on Matrix with Exponential Decay**
> >
> > **Data**: Matrix(100 $\times$ 100) with power decay $\alpha = 0.5$, $k = 100$.
> >
> > | $n_{iter}$ | $Err_{avg}(S_{rayleigh})$ | $Err_{max}(S_{rayleigh})$ | $Err_{min}(S_{rayleigh})$ | Time (sec) |
> > | ---------- | ---------------------------- | ---------------------------- | ---------------------------- | -------------------- |
> > | 1 | $6.8593 \times 10^{-5}$ | $3.6620 \times 10^{-3}$| $< 1.0000 \times 10^{-7}$    | $4.2 \times 10^{-3}$ |
> > | 3 | $6.6933 \times 10^{-5}$ | $3.1740 \times 10^{-3}$ | $< 1.0000 \times 10^{-7}$    | $5.5 \times 10^{-3}$ |
> > | 5 | $5.8387 \times 10^{-5}$ | $3.1740 \times 10^{-3}$ | $< 1.0000 \times 10^{-7}$    | $7.3 \times 10^{-3}$ |
> >
> > We can observe that the average estimation error is approximately within $1 \times 10 ^{-4}$. Next, we evaluate the SVD performance based on $R_{\text{acc}}$ for different estimation accuracies at this error level. Here, the estimation error is artificially introduced by adding Gaussian noise to the estimated singular value. This is based on our observation that such noise has a uniform effect on singular values, regardless of their magnitude. Specifically, we define the singular value matrix with Gaussian noise as $S_{noise}(b) = S + bE$, where each component $E_{ij} \sim N(0,1)$ represents Gaussian noise. The results show that our method exhibits robust performance against estimation error.
> >
> > **Table 4: The Performance of Singular Value Estimation under Different Noise Levels**
> > | Data | m,n | k  | b = 0 | b = $1 \times 10^{-5}$ | b = $1 \times 10^{-4}$ | b = $1 \times 10^{-3}$ |
> > | ----------------------------------------- | --------- | --- | ------ | ---------------------- | ---------------------- | ---------------------- |
> > | Baboon.png   | 256,256   | 100 | 0.9072 | 0.9072 | 0.90710| 0.9003 |
> > | Hill.png| 512,512   | 100 | 0.9612 | 0.9611  | 0.9578 | 0.9511 |
> > | Matrix with power decay $\alpha=0.5$ | 100,100   | 100 | 0.9979 | 0.9978| 0.9978| 0.9965 |
> > | Matrix with exponential decay $\beta=0.5$ | 1000,1000 | 250 | 0.9999 | 0.9999 | 0.9999 | 0.9998|

---

### Author Response · Authors · 2025-11-25
**General Response**

We sincerely thank all reviewers for their careful reading and constructive feedback. We are encouraged that reviewers highlighted the following strengths: originality and novelty (nBDS, 6HC2, bebZ), clear and easy-to-follow writing (ZVWz, bebZ), relevance to current research directions (ZVWz), strong empirical performance (6HC2), and sound problem motivation and significance (6HC2, bebZ).

In this rebuttal, we address the concerns with additional experiments and clarifications, summarized below.

**Clarifications:**
- **Conceptual clarification**
  We clarify the distinction between “descent methods” and “matrix-based methods” (Reviewer 6HC2 W1). Regarding the advantage over matrix-based methods, we replace the earlier broad claim with a more precise and balanced statement about when matrix-based approaches can be less parallel-friendly (Reviewer 6HC2 W4).

- **The role of singular value estimation**
  We explain that singular value estimation does not mean the SVD has been solved (Reviewer 6HC2 W2). We show that singular value estimation and other pre-processing (e.g., the calculation of the compatible matrix) require only a negligible amount of additional computation (Reviewer ZVWz Q4; Reviewer 6HC2 W2; Reviewer bebZ Q2), which is supported by additional experiments (refer to the **Experiments** section).

- **Algorithm clarifications**
  We add a failure-detection and auto-restart mechanism to improve robustness (Reviewer ZVWz Q3; Reviewer bebZ Q5). We clarify why performing normalization once at the end is correct in our algorithm (Reviewer ZVWz Q3; Reviewer bebZ W2). We also report hyperparameters and implementation details for Rayleigh iteration, randomized subspace iteration, and line-search strategies to improve reproducibility (Reviewer bebZ Q6).

- **Convergence guarantees and robustness analysis**
  We expand the analysis to include convergence guarantees and a robustness discussion, completing the theoretical picture of the method under practical conditions (Reviewer 6HC2 W3; Reviewer bebZ W4).

- **Notation and wording**
  We fix typos (Reviewer ZVWz Q1; Reviewer 6HC2 Q1) and refine derivations for precision and clarity (Reviewer ZVWz Q2).

**Experiments:**
- **Computational cost breakdown and parallelization rationale**
  We provide a breakdown showing that estimating singular values and forming the compatible matrix $D$ consume only a small fraction of runtime, supporting the practicality of parallelization (Reviewer ZVWz Q4; Reviewer 6HC2 W2; Reviewer bebZ Q2). We also compare sequential and parallel implementations to demonstrate efficiency gains (Reviewer 6HC2 W4; Reviewer bebZ W3).

- **Ablations for Newton-method parameters**
  We conduct systematic ablations on the Newton-step hyperparameters to assess sensitivity and provide robust default settings (Reviewer bebZ Q3).

- **Robustness of the singular value estimation step**
  We add robustness experiments highlighting the efficiency and reliability of the Rayleigh iteration used in the singular-value estimation stage (Reviewer ZVWz Q4; Reviewer 6HC2 W3; Reviewer bebZ Q1).

- **Applicability for larger scales**
  We report results on large-scale random matrices ($10{,}000 \times 10{,}000$) to demonstrate the scalability and broad applicability of Des-SVD (Reviewer nBDS W3).

- **Baseline selection and justification**
  To justify baseline selection, we compare Lan-SVD with LOBPCG (Reviewer nBDS W1) and with Power Iteration (Reviewer bebZ W3), illustrating respective strengths in sparse and dense settings. This clarifies our experimental design and supports fair comparisons.

We hope these additions will address the reviewers’ concerns and strengthen our contribution. We are preparing the revised PDF and sincerely look forward to the reviewers’ valuable feedback, with endorsed points to be reflected in the manuscript.

---

### Author Response · Authors · 2025-11-28
**Updated Manuscript Draft and Request for Further Feedback**

Dear Reviewers and Area Chairs,

Over the past few days, we have carefully revised the manuscript and prepared an updated PDF version that incorporates the valuable comments from the first review round. All modified content has been highlighted in blue for your convenience. Please note that this version is still a working draft. We would be sincerely grateful for any further feedback or suggestions you may have, so that we can reflect our discussions in the next revision of the manuscript.

Thank you very much for your time and consideration.

Best regards,
Authors of Paper 7072

---

### Author Response · Authors · 2025-12-02
**Recap For Area Chair**

Dear ACs, SACs, and PCs,

We sincerely appreciate the reviewers’ time, effort, and constructive feedback during the first round of the rebuttal.


In our work, we present the first practically usable descent-based algorithm for computing the SVD in decades — a contribution acknowledged by all reviewers in terms of novelty, quality, and significance. For instance, Reviewer 6HC2 (initial score: 2) commended the paper for its *" novel theoretical insight"*, *"strong empirical performance"*, and *"valid problem motivation"*. Similarly, Reviewer bebZ (initial score: 4) praised the paper for its *"a clear primal–dual route to SVD"*, noting that *"theoretical results are clear and aligned with algorithmic design"*, and emphasized its relevance for *"large-scale, distributed, or privacy-conscious settings where matrix factorization pipelines are awkward"*. Furthermore, they observed that the work *" addresses an important gap"* and *"could influence future work"*.

The main concerns raised in the initial reviews focus on two points:

### 1. “Descent Method” vs. “Matrix-Based Method”.

This issue was raised primarily by Reviewer 6HC2, and it is indeed an important conceptual question. We believe the distinction is clear, as Newton’s method is widely recognized as a gradient-based descent method that identifies a descent direction and employs line-search or damping. Thus, Des-SVD fits naturally within the descent framework.

### 2. Prior Singular Value Estimation

This concern was shared among several reviewers. Reviewer 6HC2 interpreted this step as implying that the “core SVD problem is assumed solved,” while Reviewers ZVWz and bebZ asked about its computational details, including robustness, step-size choices, and whether the assumptions required by our theory are satisfied. During the discussion stage, we additionally reported:

- its time cost (only about $1$% of the total runtime)

- the typical estimation error (within $1 \times 10^{-4}$)

- experiments under random errors (up to $1 \times 10^{-3}$)

We hope the additional experiments could clearly clarify: the prior singular value estimation takes relatively negligible computational time, and Des-SVD is robust to the estimation error.

Finally, we appreciate your effort in evaluating our submission under this year’s extraordinary review circumstances. All revisions are included in the updated PDF, with changes highlighted in blue for your convenience.

---

### Meta-Review · Area_Chair_2Hyg · 2026-01-02

**Summary:**

After reviewing the referee comments, the authors’ responses, and considering my own evaluation, I believe the manuscript has merit but still falls short in several key aspects:

1.	As noted by the reviewers, the experimental design does not adequately support the claims regarding parallelization and distributed computation. The additional experiments in Section 5.4 lack comparative baselines and are therefore not convincing.

2.	While the authors argue in their rebuttal that their method may offer advantages in privacy sensitive scenarios, the work would be more persuasive if the method were formally formulated and evaluated within a federated learning framework.

3.	As shown in Table 3, the proposed Des-SVD does not outperform Lan-SVD. Moreover, like Lan-SVD, it still requires users to pre-specify the parameter $k$. Thus, Des-SVD does not appear to represent a substantial advance over the current state of the art in SVD methods.

4.	The proposed categorization of SVD methods into matrix based and gradient based approaches seems unnecessary and controversial.

**Reviewer Scores:**

NA

---

### Decision · Program_Chairs · 2026-01-26

Reject